# Dynamics of GLP-1R peptide agonist engagement are correlated with kinetics of G protein activation

Giuseppe Deganutti [1,2,11], Yi-Lynn Liang[3,8,11], Xin Zhang[3,4,11], Maryam Khoshouei[5,9,11], Lachlan Clydesdale[3,11], Matthew J. Belousoff[3,4], Hari Venugopal[6], Tin T. Truong[3], Alisa Glukhova [3,10], Andrew N. Keller[3], Karen J. Gregory [3], Katie Leach [3], Arthur Christopoulos [3,4], Radostin Danev [7], Christopher A. Reynolds [1,2✉], Peishen Zhao [3,4✉], Patrick M. Sexton [3,4✉] & Denise Wootten [3,4✉]

The glucagon-like peptide-1 receptor (GLP-1R) has broad physiological roles and is a validated target for treatment of metabolic disorders. Despite recent advances in GLP-1R structure elucidation, detailed mechanistic understanding of how different peptides generate profound differences in G protein-mediated signalling is still lacking. Here we combine cryo-electron microscopy, molecular dynamics simulations, receptor mutagenesis and pharmacological assays, to interrogate the mechanism and consequences of GLP-1R binding to four peptide agonists; glucagon-like peptide-1, oxyntomodulin, exendin-4 and exendin-P5. These data reveal that distinctions in peptide N-terminal interactions and dynamics with the GLP-1R transmembrane domain are reciprocally associated with differences in the allosteric coupling to G proteins. In particular, transient interactions with residues at the base of the binding cavity correlate with enhanced kinetics for G protein activation, providing a rationale for differences in G protein-mediated signalling efficacy from distinct agonists.

[1] Centre for Sport, Exercise and Life Sciences, Coventry University, CV1 5FB Coventry, UK. [2] School of Biological Sciences, University of Essex, Colchester CO4 3SQ, UK. [3] Drug Discovery Biology, Monash Institute of Pharmaceutical Sciences, Monash University, Parkville, VIC 3052, Australia. [4] ARC Centre for Cryo-electron Microscopy of Membrane Proteins, Monash Institute of Pharmaceutical Sciences, Monash University, Parkville, VIC 3052, Australia. [5] Department of Molecular Structural Biology, Max Planck Institute of Biochemistry, 82152 Martinsried, Germany. [6] Ramaciotti Centre for Cryo-Electron Microscopy, Monash University, Clayton, VIC 3168, Australia. [7] Graduate School of Medicine, University of Tokyo, N415, 7-3-1 Hongo, Bunkyo-ku, Tokyo 113-0033, Japan. [8] Present address: Confo Therapeutics, Technologiepark 94, Ghent (Zwijnaarde) 9052, Belgium. [9] Present address: Novartis Institutes for Biomedical Research, Novartis Pharma AG, Basel, Switzerland. [10] Present address: Walter and Eliza Hall Institute, Parkville, VIC 3052, Australia. [11] These authors contributed equally: Giuseppe Deganutti, Yi-Lynn Liang, Xin Zhang, Maryam Khoshouei, Lachlan Clydesdale. ✉email: ad5291@coventry.ac.uk; elva.zhao@monash.edu; patrick.sexton@monash.edu; denise.wootten@monash.edu

The glucagon-like peptide-1 receptor (GLP-1R) is widely expressed in many tissues and mediates the action of the gastrointestinal peptide hormone, glucagon-like peptide-1 (GLP-1)[1]. GLP-1 has numerous physiological effects that are desirable in the management of type 2 diabetes and obesity, including regulation of insulin secretion, slowing gastric emptying, suppressing appetite and regulating carbohydrate metabolism. Numerous endogenous agonists activate the GLP-1R, including several forms of GLP-1, oxyntomodulin and glucagon, and multiple exogenous peptide agonists are approved, or in clinical development, for the treatment of type 2 diabetes and/or obesity[1,2]. However, these have different therapeutic efficacies for glucose control, weight loss, improved cardiovascular outcomes, as well as side effect profiles, such as nausea and vomiting[2]. These differential effects may be attributed to their pharmacokinetic profiles and/or distinctions in how each peptide binds and activates the GLP-1R.

The GLP-1R is a class B1 G protein-coupled receptor (GPCR) that mediates its effects via coupling to heterotrimeric G proteins[1]. The receptor is predominantly coupled to the stimulatory G protein $G_s$ to raise cAMP levels within the cell, however, it pleiotropically couples to multiple transducers, including other G protein subtypes and regulatory proteins[1,3]. When compared to GLP-1, other GLP-1R agonists can display differential efficacies within a single signalling pathway, as well as preferential signalling towards individual pathways at the expense of others[4–8]. These phenomena lead to biased agonism, which is commonly observed when GLP-1R agonists are assessed across multiple signalling pathways. However, the molecular basis for how individual agonists can promote profound differences in pharmacology is still poorly understood.

Class B1 GPCRs bind their peptide agonists via a two-domain model, whereby the C-terminus of the peptide interacts with the receptor extracellular N-terminal domain (ECD) promoting an "affinity trap" that enables the engagement of the N-terminus of the peptide with the receptor transmembrane domain (TMD), with interactions with the TMD required for receptor activation[9]. In recent years, advances in cryo-electron microscopy (cryo-EM) have enabled structural determination of a large number of class B1 GPCRs bound to their endogenous agonists, and coupled to $G_s$, including that of the GLP-1R, which confirm engagement of these peptides with both the ECD and the TMD[10–21].

Naturally occurring GLP-1R peptide agonists share a conserved N-terminal sequence with GLP-1, including oxyntomodulin and the first FDA approved GLP-1R agonist, exendin-4 (Fig. 1a). Despite this conservation, these peptides induce distinct signalling profiles and have different mechanisms for receptor interaction[4]. Truncation of just two N-terminal residues of GLP-1 decreases affinity by 100–300-fold and potency for cAMP signalling by 1000–10,000-fold[22]. In contrast, truncation of the first two residues of exendin-4 does not significantly alter its affinity but does lower potency for cAMP signalling by ~200-fold[23–25]. Unlike naturally occurring GLP-1R agonists, exendin-P5 contains a unique N-terminal sequence (Fig. 1a). While this peptide is a potent agonist for cAMP production, it is a biased agonist, with a preference for G protein-mediated signalling relative to β-arrestin recruitment when compared to GLP-1 and exendin-4[26]. Intriguingly, this peptide also displays a unique in vivo profile relative to exendin-4 with an improved ability to reduce hyperglycaemia in animal models of diabetes.

Due to the therapeutic implications of biased agonism and differential efficacy, understanding the molecular details of how different peptides engage and activate the GLP-1R is crucial. Structures of the GLP-1R in complex with $G_s$, bound with GLP-1 and exendin-P5, as well as non-peptide ligands, have been determined using cryo-EM[13,16,27]. Coupled with mutagenesis data, these structures provide initial molecular insights into biased agonism and differential efficacy. The conformation of TM6-ECL3-TM7 has been correlated with the distinct signalling profiles of GLP-1 and exendin-P5, and along with the conformation of TM1, TM2, and ECL2, are key receptor domains important for GLP-1R cAMP signalling, and biased agonism[3,13,28,29]. However, a detailed mechanistic understanding of GLP-1R activation linked to downstream signalling is still lacking, with the existing structural information unable to fully explain how differential peptide efficacies arise, or the differential requirements of the N-terminus and C-terminal sequences of peptide analogues.

In this work, we investigate the molecular mechanisms by which GLP-1, oxyntomodulin, exendin-4, and exendin-P5 bind and activate the GLP-1R using a combination of structural biology, molecular dynamics simulations, and pharmacological studies combined with extensive receptor mutagenesis. We show that different peptides engage with residues in the receptor TM domain with different dynamics that are correlated with differences in the allosteric communication between the peptide and G protein-binding sites.

## Results

**Cryo-EM determination of the GLP-1R:$G_s$ complex bound by oxyntomodulin and exendin-4.** Cryo-EM structures of exendin-4- and oxyntomodulin-bound GLP-1R-$G_s$ complexes were determined using established methodology[12–14]. Purified complexes (Supplementary Fig. 1) that contained all expected components were vitrified and imaged by single-particle cryo-EM on a 300 kV Titan Krios, with (oxyntomodulin) or without (exendin-4) a Volta Phase Plate (VPP). Processing these datasets yielded consensus maps with global resolutions of 3.3 Å (oxyntomodulin) and 3.7 Å (exendin-4) at gold standard FSC 0.143 (Fig. 1b, c, Supplementary Fig. 1). As there was only limited density for the α-helical domain (AHD) of the Gα subunit, this was masked out during the refinement. Similar to the previous peptide-bound:GLP-1R:$G_s$ complex structures, the highest resolution was observed within the G protein and receptor TMD, with lower resolution in the extracellular half of the receptors, including the ECD (Supplementary Fig. 1), indicative of greater flexibility in these regions.

The cryo-EM map for the oxyntomodulin-bound complex enabled robust modelling and confident assignment of most of the side-chain rotamers for oxyntomodulin, the G protein and the receptor TMD (Supplementary Fig. 2a), with the exception of the extracellular loop (ECL) 1 and intracellular loop (ICL) 3, which were not modelled. The ECD was less well resolved, however, the protein backbone could be ab initio modelled into the density. The exendin-4-bound complex had lower global resolution however, robust modelling into the map could be performed for the majority of the peptide, the G protein and receptor TMD (Supplementary Fig. 2b). ECL1, ECL3 and ICL3 were not modelled as the density was less well resolved indicative of higher flexibility within these domains. The low resolution within the ECD of the exendin-4-bound map precluded confident modelling; as such the ECD was rigid body fitted to the density, followed by MD refinement of the backbone.

**General features of peptide bound GLP-1R:$G_s$ complexes.** The oxyntomodulin and exendin-4 bound GLP-1R:$G_s$ complexes exhibited key features of active state class B1 GPCRs (Supplementary Fig. 3) and are consistent with the general features of the active state GLP-1R observed previously when bound by other agonists[13,16,27]. Relative to the inactive GLP-1R[30], this includes an upwards and clockwise rotation of the ECD relative to the TMD, a reorganisation of the extracellular TM regions to

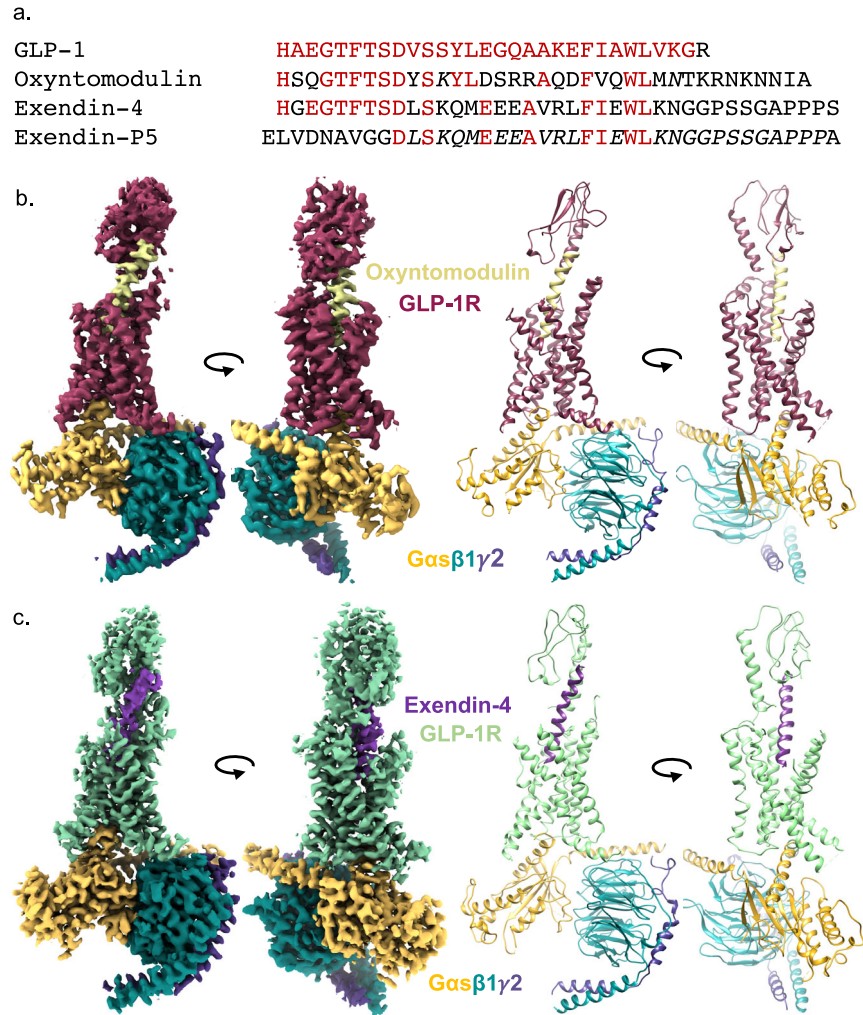

a.

| GLP-1 | HAEGTFTSDVSSYLEGQAAKEFIAWLVKGR |
| Oxyntomodulin | HSQGTFTSDYSKYLDSRRAQDFVQWLMNTKRNKNNIA |
| Exendin-4 | HGEGTFTSDLSKQMEEEAVRLFIEWLKNGGPSSGAPPPS |
| Exendin-P5 | ELVDNAVGGDLSKQMEEEAVRLFIEWLKNGGPSSGAPPPA |

**Fig. 1 Cryo-EM structures of GLP-1R:G$_s$ complexes with different agonists. a** Sequence of peptides was assessed in this study. **b** and **c** orthogonal views of the cryo-EM maps (left), and the backbone models built into the maps in ribbon format (right) for oxyntomodulin (**b**) and exendin-4 (**c**) bound GLP-1R:G$_s$ complexes. Colouring denotes the protein segments as highlighted on the figure panels.

accommodate peptide engagement and rearrangement of a conserved central polar network to stabilise a sharp kink within the centre of TM6, which facilitates the large outward movement of this TM at the intracellular face that is required to accommodate G protein binding. Similar to other peptide-bound GLP-1R structures[13,16], exendin-4 and oxyntomodulin adopted a continuous alpha helix, with their C-terminus bound within the ECD and their N-terminus bound deep within the TMD forming extensive interactions with residues within TM1, TM2, TM3 TM5, TM6, TM7 and ECL2 (Figs. 1, 2, Supplementary Fig. 4).

Comparison of the oxyntomodulin-bound structure with the previously determined high-resolution GLP-1-bound GLP-1R (PDB 6X18)[13,16] reveals remarkably similar TMD conformations, with only a small difference in position of the ECD relative to the bundle (Fig. 3a). In addition, the side-chain rotamers within the TMD cavity were also similar, albeit the strength and nature of their interactions with the bound peptides vary (Fig. 3b). Exendin-4 also engages the GLP-1R in a comparable manner, with the ECD adopting a similar conformation to the GLP-1-bound receptor (Fig. 3a). However, within the TMD, the exendin-4 bound cavity is more open, predominantly due to a more outward location of TM1, however, TM2, TM4-ECL2-TM5 and TM7 are also located further from the centre of the bundle (Fig. 3b). Nonetheless, the receptor side chains within the pocket exhibit similar rotamers. Comparison of the exendin-4 and

exendin-P5 bound (PDB 6B3J) structures revealed similarities in their ECD location and greater similarities of the backbone orientations for TMs 1–5, relative to the GLP-1-bound and oxyntomodulin-bound complexes (Fig. 3). However, the TMD-binding cavity is even more open in the presence of exendin-P5, due to a more outward location of TM7. While the top of TM6 and ECL3 could not be confidently modelled for the exendin-4 complex, the portion of TM6 and TM7 that were modelled, along with the weak density corresponding to ECL3 supports a backbone conformation more similar to GLP-1 and oxyntomodulin bound receptors, rather than exendin-P5, albeit it is likely that this region is more conformationally dynamic.

**Cryo-EM structures, molecular dynamics and mutagenesis reveal distinct dynamic interactions of individual peptides within the GLP-1R binding site.** Atomic modelling into the static consensus cryo-EM maps revealed specific details regarding the interactions of oxyntomodulin and exendin-4 with the GLP-1R. These are reported in Supplementary Table 1 and the interactions of the peptide N-termini with the TMD are shown in Fig. 2. The peptide N-terminus is highly conserved between GLP-1, exendin-4 and oxyntomodulin, and as such, a large number of receptor contacts are also conserved, whereas these are more divergent when compared with exendin-P5 (Supplementary

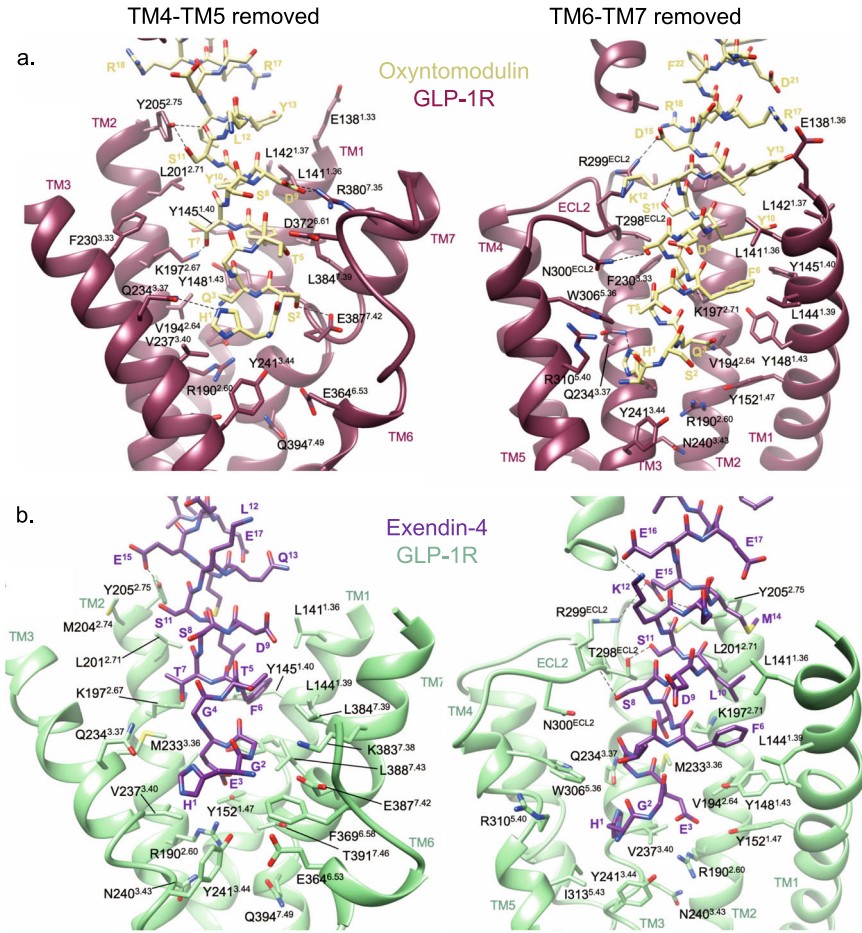

**Fig. 2 Interactions of oxyntomodulin and exendin-4 peptides within the TMD binding cavity of the GLP-1R. a** GLP-1R (dark pink) and oxyntomodulin peptide (pale yellow). **b** GLP-1R (pale green) and exendin-4 (purple). For each binding site two views are depicted for clarity; Left, side view of the TM bundle viewed from the upper portion of TM4/TM5 where TM4-ECL2-TM5 have been removed; Right; side view of the TM bundle viewed from the upper portion of TM6/TM7 where TM6-ECL3-TM7 has been removed. Dashed lines depict hydrogen bonds as determined using default settings in UCSF chimera. Superscript numbering for receptor residues refers to the generic Wootten et al. class B1 numbering system[40].

Table 1). To interrogate the relative importance of consensus structure interactions for receptor binding and activation, we employed receptor mutagenesis, where each residue within the TMD that formed an interaction with any of the four peptides in the static cryo-EM structures was mutated to alanine (with the exception of A368$^{6.57}$, which was mutated to glycine), and the binding affinity and cAMP signalling of each peptide was assessed. From concentration-response curves (Supplementary Figs. 5–8), pIC$_{50}$ values, and transduction ratios (log $\tau_c/K_A$, where receptor expression was also taken into account) that quantify signalling efficiency, were calculated. These were compared between the mutant and wild-type receptors to assess the impact of the mutation on affinity and signalling of each peptide (Supplementary Figs. 9, 10, Supplementary Table 2), and these were mapped onto the cryo-EM structures (Fig. 4). When comparing the global mutagenesis profile, the effects on exendin-4 were similar to those on GLP-1, with a strong positive correlation observed for the mutagenesis data for both affinity and cAMP production, albeit that the effect on cAMP signalling was generally smaller for exendin-4 (Figs. 4 and 5). Interestingly, while oxyntomodulin-bound GLP-1R displayed a more similar TMD conformation to that bound to GLP-1 in the static cryo-EM structure, the effect of mutagenesis was more divergent, with numerous mutations differentially impacting oxyntomodulin affinity and/or signalling data relative to GLP-1 (Fig. 5,

Supplementary Figs. 9, 10, Supplementary Table 2). Nonetheless, there was still a significant weak positive correlation across all mutant datasets and similar to exendin-4, mutations affecting both peptides generally had a greater effect on GLP-1 than oxyntomodulin signalling (Figs. 4, 5). In contrast, the exendin-P5 mutagenesis profile was very distinct from the other peptides with few mutations altering exendin-P5 affinity and only a very weak, albeit significant, correlation with the effect of mutations on GLP-1 in cAMP signalling assays (Figs. 4, 5, Supplementary Figs. 9, 10). In addition, there was no correlation between the oxyntomodulin and exendin-P5 mutagenesis when assessing the effect of mutations as a whole, albeit there were select mutations that exhibited similar effects on the signalling of both peptides (Figs. 4, 5, Supplementary Figs. 9, 10).

The divergence in the effect of mutagenesis of residues comprising the TM-binding pocket, even where interactions in the static consensus structures were similar, confirmed that static visualisation of complexes is insufficient to fully understand binding and activation mechanisms, and suggests that the dynamics of peptide-receptor engagement likely play a critical role. Therefore, we probed the stability and dynamics of each receptor complex in a simulated POPC lipid environment over microseconds of molecular dynamics (MD) simulations (Supplementary Movie 1). Regions that were not modelled in the cryo-EM maps due to the low resolution were first modelled and the

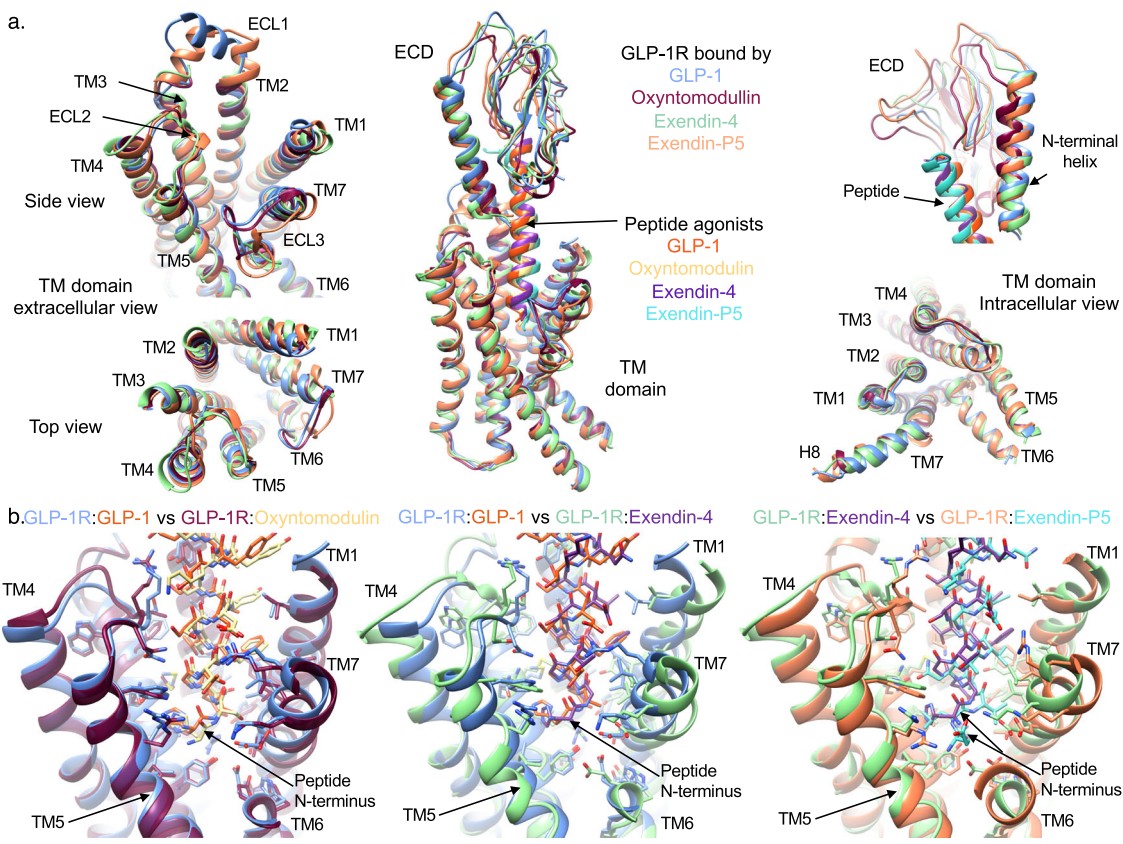

**Fig. 3 Comparisons of the GLP-1R conformations and binding pockets stabilised by GLP-1, exendin-4, oxyntomodulin and exendin-P5.**
**a** Superimposition of the receptor from the GLP-1R:Gs complex structures bound with GLP-1 (6X18[16]—receptor-blue, peptide-orange), oxyntomodulin (receptor-dark pink, peptide-pale yellow), exendin-4 (receptor-pale green, peptide-purple) and exendin-P5 (6B3J[13]—receptor-pale orange, peptide-cyan). Middle, Overlay of full-length receptors with bound peptides; Left, close up of an extracellular portion of the receptor TMD viewed from the side (top) and looking down on the TMD-binding cavity (bottom); Right, close up of the ECD showing the distinct location of the ECD N-terminal α-helix and the location of the peptide C-terminus in the different structures (top) and the receptor TM domain viewed from the intracellular G protein binding site.
**b** Superimposition of the peptide binding sites within the GLP-1R TMD comparing GLP-1 with oxyntomodulin (left), GLP-1 with exendin-4 (middle) and exendin-4 with exendin-P5 (right). Colouring denotes the different peptide bound receptors as highlighted on the figure panels.

receptor complex energy minimised, prior to commencing the simulations. The main peptide–receptor interactions identified in the MD are summarised in Fig. 6, Supplementary Fig. 11, and Supplementary Table 3. The MD studies revealed that very few residues, particularly within the TMD, formed stable peptide contacts, instead these interactions were transient (Supplementary Fig. 11). Nonetheless, there are common residues within the ECD, TM1, TM2, ECL2, TM5, TM6 and TM7 that interact with all four peptides, albeit the nature and stability of these interactions differed considerably between the agonists. In contrast, more divergent interaction patterns were observed within the TMD-ECD linker region (stalk), ECL1, TM2 and ECL3/TM7 (Supplementary Fig. 11).

Consistent with the cryo-EM structure, the MD analysis revealed the GLP-1 N-terminus forms extensive interactions with TMs 1, 2, 3, 5, 6, 7 and ECLs 1–3 (Fig. 6, Supplementary Fig. 11). The interactions of oxyntomodulin differed, with more transient contacts deep within the peptide binding cavity and with residues located higher within TM2, and with ECL2. This was coupled with enhanced interactions at the top of TM1 and distinct and more sustained interactions within TM7 (Fig. 6, Supplementary Table 3). Within the N-terminal nine amino acids, oxyntomodulin differs from GLP-1 by 2 residues, with Ala8 and Glu9 of GLP-1 replaced by Ser2 and Gln3 in oxyntomodulin (Fig. 1a). These residues are located at the base of the GLP-1R binding pocket in the cryo-EM structures and interact with residues in

TMs 1, 2 and 7 (Figs. 2 and 3, Supplementary Fig. 4, Supplementary Table 3). While Glu9 of GLP-1 formed strong and persistent hydrogen bond and van der Waals interactions with R190[2.60], Y152[1.47], and to a lesser extent, Y148[1.43], the polar side chain of Gln3 in oxyntomodulin only formed weak and very transient hydrogen bonds with these residues, and van der Waals interactions only with Y148[1.43] (Movie S1, Supplementary Table 3). Consistent with this, alanine substitution of Y148[1.43] had a similar influence on the affinity of both ligands, while Y152[1.47]A had a greater effect on GLP-1 affinity, and R190[2.60]A decreased GLP-1 affinity >30-fold, but oxyntomodulin was unaffected (Supplementary Figs. 5 and 9, Supplementary Table 2). All three residues play a much greater role in GLP-1 mediated cAMP signalling, relative to oxyntomodulin (Figs. 4 and 5, Supplementary Figs. 7 and 10). In addition to these interactions, Gln3 of oxyntomodulin formed transient hydrogen bonds and hydrophobic contacts with T391[7.46] and E387[7.42], as well as van der Waals interactions with L388[7.43], while Glu9 in GLP-1 only formed hydrophobic interactions with these residues. While Ser2 of oxyntomodulin and Ala8 of GLP-1 both interacted with TM7 residues, these interactions were stronger with oxyntomodulin due to a persistent hydrogen bond between Ser2 and E387[7.42] and transient hydrogen bonds and van der Waals contacts with K383[7.38], D372[ECL3] and L384[7.39] (Supplementary Table 3). Accordingly, alanine mutagenesis of TM7 residues influenced both GLP-1 and oxyntomodulin affinity, but there was a larger

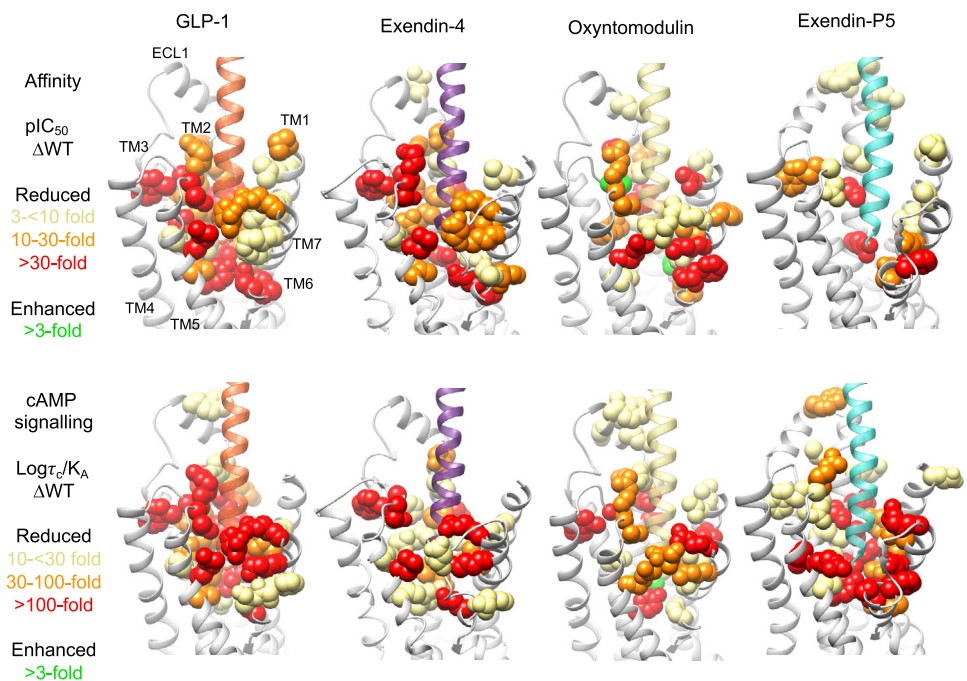

**Fig. 4 Heat maps depicting the 3D representation of the effect of alanine mutation of residues within the TMD peptide-binding cavity on affinity and signalling.** Models of the peptide bound GLP-1Rs showing residues (in space fill) that altered affinity (top) or signalling (bottom) of GLP-1, exendin-4, oxyntomodulin and exendin-P5 relative to the wild type receptor when mutated. These are coloured depending on their level of effect highlighted in the colour key. TM domains are labelled on the GLP-1-bound model depicting affinity changes, with the same receptor views used for the remaining models.

effect on oxyntomodulin, consistent with its stronger interactions, however, interestingly these residues were more important for GLP-1-mediated cAMP production (Supplementary Figs 9, 10). In both peptides, the N-terminal histidine sits in an enclosed pocket-forming hydrophobic and hydrogen bond interactions with E364$^{6.53}$, E387$^{7.42}$, R310$^{5.40}$, Y241$^{3.44}$, W306$^{5.36}$, I313$^{5.43}$ and Q234$^{3.37}$ however, these interactions are more persistent with GLP-1, and this is likely linked to the stability of interactions of Glu9 and with residues at the base of the binding cavity (Supplementary Table 3, Supplementary Movie 1).

In the MD studies, weaker interactions of oxyntomodulin with residues deep in the cavity, particularly R190$^{2.60}$, were coupled with a loss of stable interactions along the entire face of the TM2 helix and weaker interactions with ECL2, all of which interact with the same face of the helical GLP-1 peptide (Fig. 6). While mutagenesis studies clearly reveal that these residues within TM2 and ECL2 are important for both GLP-1 and oxyntomodulin function, alanine mutations in these regions generally had a larger impact on GLP-1 (Figs. 4 and 5, Supplementary Figs. 9 and 10). Overall, these differing receptor interaction patterns of GLP-1 and oxyntomodulin were associated with larger conformational dynamics within the oxyntomodulin binding pocket during the course of the MD simulation, with the cavity opening and closing, likely due to the lack of stable interactions within the base of the TMD binding pocket, TM2 and ECL2, coupled with more persistent interactions with the upper regions of TM1 and TM7, relative to GLP-1 (Fig. 6, Supplementary Fig. 12, Supplementary Movie 1).

While the C-terminus of oxyntomodulin and the receptor ECD/ECL1 was lower resolution in the cryo-EM map, the interaction patterns in the modelled protein and the MD simulations were largely similar to GLP-1, albeit that the nature of some interactions differed due to their differing sequences; residues capable of hydrogen bonding were present in one peptide, but not the other. For example, Arg17 and Arg18 of oxyntomodulin form interactions with the ECD and top of TM2

in the simulations, whereas the corresponding alanine residues in GLP-1 could not form these interactions. Interestingly the ECD was also more mobile in the oxyntomodulin bound receptor, when compared to GLP-1 (Supplementary Fig. 12), which may be associated with the greater dynamics within the bundle, given the bound peptide bridges these two domains and stabilises their motions relative to one another.

The more open TMD in the static exendin-4 bound cryo-EM structure was associated with fewer stable peptide contacts (Supplementary Table 1), relative to GLP-1 and oxyntomodulin, however, the cryo-EM map was a lower resolution, which precluded modelling of some of the binding site (top of TM6, ECL3 and ECL1). The MD simulations revealed a large number of conserved interactions between GLP-1 and exendin-4, including the majority of peptide hydrogen bonding within the TMD. Of particular note was the lack of persistent receptor interactions for His1 of exendin-4, in contrast to the extensive interactions observed for GLP-1 (His7) and oxyntomodulin (His1), as described above (Fig. 3, Supplementary Tables 1 and 3, Supplementary Movie 1). His1 of exendin-4 formed only limited transient interactions with TM5, however more persistent interactions with E364$^{6.53}$ in TM6 were evident in the MD simulations (Supplementary Table 3). Interestingly, His1-Gly2 of exendin-4 exhibited weaker density relative to the rest of the peptide in the cryo-EM map, and this is consistent with the enhanced flexibility observed for this region of the bound peptide in the simulations, likely due to Gly2, as glycine can destabilise α-helical conformations. The equivalent residue is a serine or alanine in oxyntomodulin and GLP-1 respectively, contributing to a more stable α-helical conformation, while also forming additional interactions at the base of the GLP-1R binding pocket (Figs. 2 and 3, Supplementary Table 3). In line with strong and stable interactions of His1-Ala2 of GLP-1 compared with transient interactions of His1-Gly2 in exendin-4, truncation of these two N-terminal residues reduces GLP-1 affinity by 100–300 fold, whereas for exendin-4, this has no effect, albeit, for both

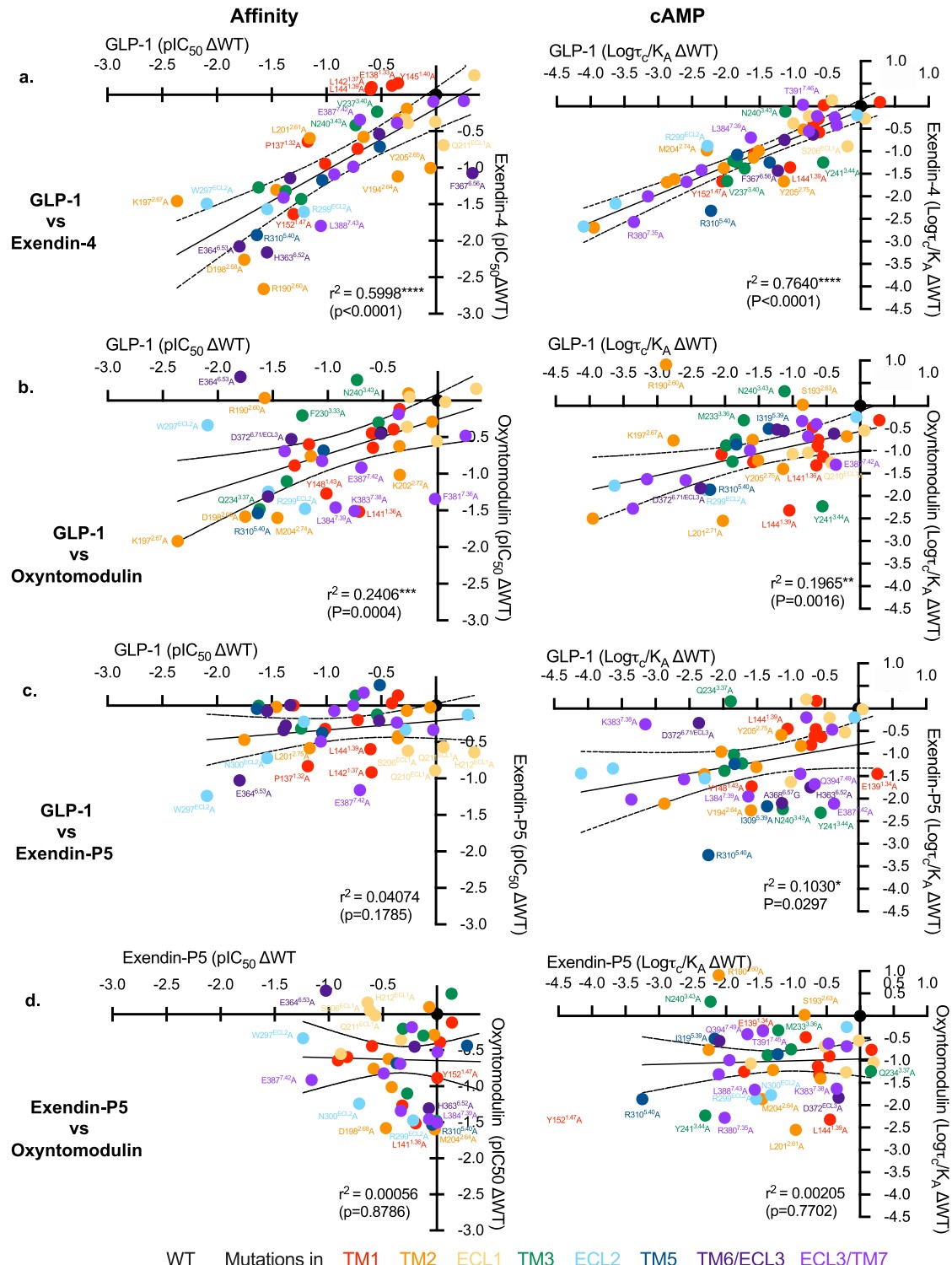

**Fig. 5 Correlation plots of the changes in peptide affinity and efficacy for TMD mutations relative to the wildtype receptor. a** GLP-1 vs. exendin-4; **b** GLP-1 vs. oxyntomodulin; **c** GLP-1 vs. Exendin-P5; **d** Exendin-P5 vs. Oxyntomodulin. Data were fit by linear regression and Pearson correlations (*r*) were determined and squared (*r*²), and the *P* value was calculated using a two-tailed critical *t* value analysis in Prism 9. The line of regression and 99% confidence intervals are displayed. Mutations are coloured relative to the receptor TM or ECL that they are located as indicated in the legend. Mutant receptors that fall outside of the 99% confidence intervals are labelled. Source data are provided in the Source Data file.

peptides, these residues are required for GLP-1R-mediated cAMP production[22–25].

Despite the flexibility within the N-terminal two residues, Glu3 of exendin-4 formed very similar contacts to that of Glu9 of GLP-1, however, interactions with residues residing at the base of the

pocket were more transient for exendin-4 (75–80% of the MD frames) than for GLP-1 in the MD simulations (97% of frames for Y152$^{1.47}$ and 100% for R190$^{2.60}$) (Supplementary Table 3). Receptor interactions for the remaining residues within the N-terminal 11 amino acids of exendin-4 were similar to those

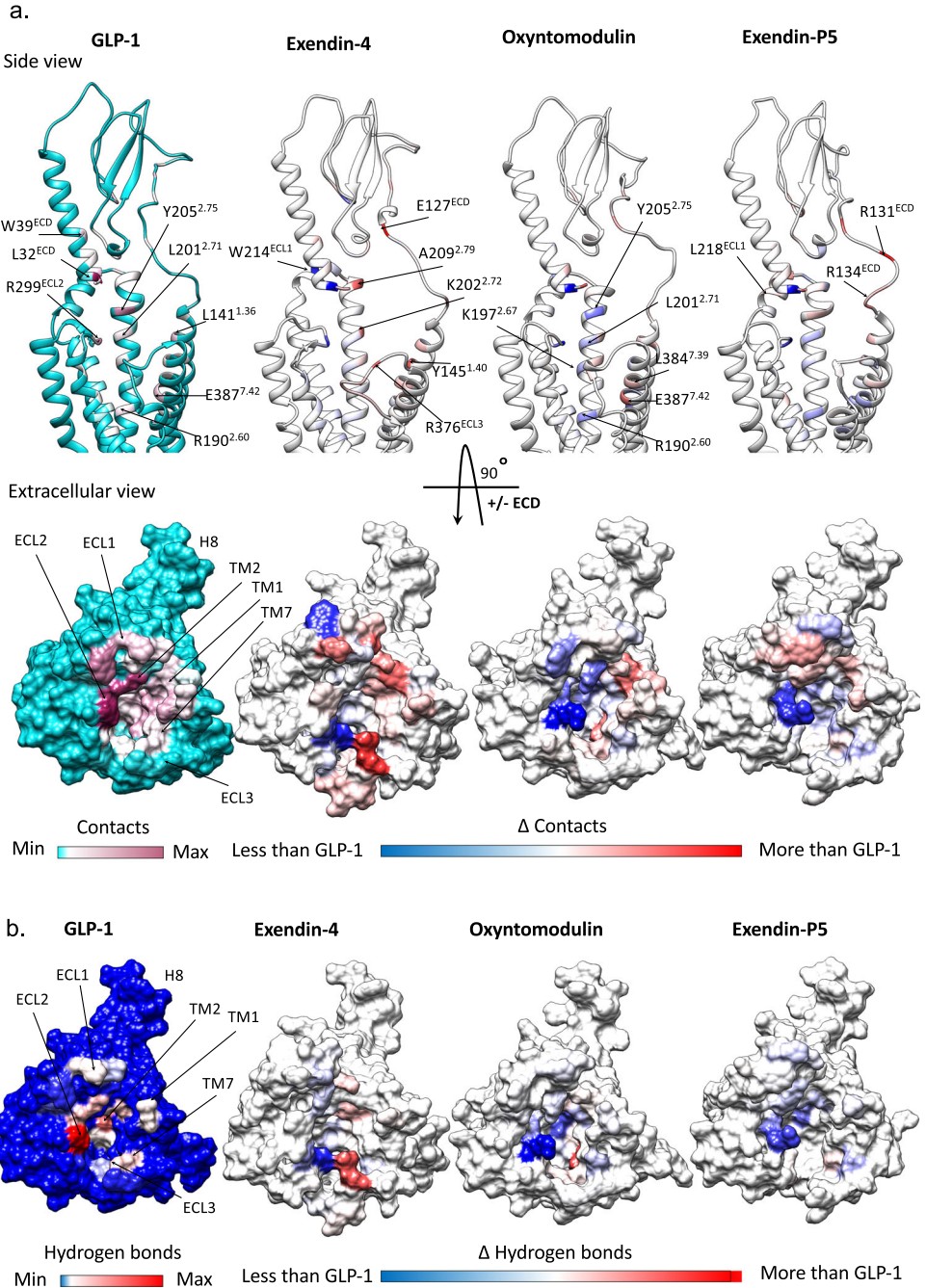

**Fig. 6 Contact differences in GLP-1R interactions of GLP-1, exendin-4, oxyntomodulin and exendin-P5 from MD simulations. a** Top; contact differences of each peptide with the GLP-1R TMD and ECD plotted on the receptor ribbon viewed from the side, Bottom; contact differences of each peptide with the TMD plotted on the GLP-1R surface viewed from the extracellular side (ECD not shown for clarity). The first column shows the GLP-1R contacts formed by GLP-1 during the simulations with no/min contacts in cyan and contacts heat-mapped from white to dark pink with increasing number/occupancy of interactions. The other three columns report the contacts differences for each residue of the GLP-1R during the MD simulation performed in the presence of other agonists with white indicating similar interactions to GLP-1, blue decreased contacts and red increased contacts. **b** Hydrogen bond differences between the four peptides plotted on the surface of the TMD viewed from the extracellular side with the ECD removed for clarity. The first column shows the GLP-1R residues involved in hydrogen bonds with GLP-1 during the MD simulations with a blue to red heatmap indicating the relative extent of interaction for each residue. The other three columns report the hydrogen bond differences for each GLP-1R residue during the MD performed in the presence of the other agonists with blue indicating fewer interactions, white similar and red more interactions, compared to GLP-1.

formed by GLP-1, albeit overall interactions deeper within TM2, TM3, TM5, and ECL2 were generally less persistent, and those in TM1 and TM7 were more persistent (Fig. 6, Supplementary Table 3, Supplementary Movie 1). The similarity in their TMD interaction patterns is consistent with the strong correlation in the impact of TMD mutagenesis for these two agonists (Fig. 5).

Moreover, the more transient interactions of exendin-4 parallel the smaller effects of the mutagenesis on cAMP signalling for this peptide.

With the exception of His1, the largest difference in the interaction of GLP-1 and exendin-4 occurred within the mid-region of the peptides where their sequences differ substantially ($E^{16}EAVRL^{21}$ for

exendin-4 vs G[22]QAAKE[27] for GLP-1). While the mid-regions of both peptides interact with the top of TM1/ECD stalk, the ECD, ECL1 and the top of TM2, exendin-4 exhibits more persistent interactions in the simulations, particularly with the TM1/stalk and ECD (Supplementary Tables 1 and 3). This, in part, may account for the higher affinity of exendin-4 for the isolated receptor ECD and the higher affinity of amino terminally truncated exendin-4 peptides, where exendin(9–39) displays only 10–30-fold lower affinity compared to exendin-4, relative to the equivalent GLP-1(15–36), which has >300-fold lower affinity than GLP-1[22,31]. These interactions may also influence the conformation of TM1 accounting for the more outward conformation in the exendin-4-bound structure (Fig. 3). Interestingly, single alanine amino acid substitutions of some interacting residues within the TMD had much larger effects on exendin-4 affinity than removal of the first 8 residues of the peptide (exendin (9–39)[22,31]), suggesting that peptide ECD and TMD interactions are correlated; non-optimal interactions of exendin-4 with the TMD elicited by receptor alanine mutations, likely promotes faster peptide dissociation from the ECD, compared to when the TMD interacting residues are not present in the peptide. This is also supported by previous studies, whereby Gly2Ala in exendin-4 was tolerated, but the converse for GLP-1 (Ala2Gly) reduced affinity[31–33]. However, the substitution of residues Glu22, Glu27 and Ala30 of GLP-1 with Gly16, Leu21 and Glu24 of exendin-4 enabled Gly2 tolerance[32].

MD simulations on the exendin-P5-bound complex revealed a similar C-terminal interaction pattern to exendin-4 for this peptide, with more persistent interactions with the ECD and TM1/stalk than GLP-1 (Fig. 6, Supplementary Fig. 11, Supplementary Table 3). Nonetheless, the TMD conformation differed and, overall, the receptor was relatively stable, exhibiting less flexibility compared with GLP-1R bound to exendin-4 or oxyntomodulin (Supplementary Fig. 12). The N-terminal sequence of exendin-P5 differs considerably from the other peptides and is also extended by one residue (Fig. 1a). Nonetheless, there are some commonalities with the other peptides in their pattern of interaction with the TMD. Val3-Asp4 interact with similar residues at the base of the binding cavity to those observed for residues 2 and 3 of the other peptides, however, with the exception of Y148[1.43], these were very transient (Supplementary Table 3). Consequently, Asp4 could also form interactions with K197[2.67] located higher within TM2. This provides a rationale for the effect of mutagenesis of R190[2.60] and K197[2.67], which were two of the few TMD residues that had a large impact on exendin-P5 affinity (Supplementary Figs 5 and 9). Despite occupying a different location in static structures, Glu1 of exendin-P5 interacts with multiple residues that interact with His1 of the other peptides, including E364[6.52], E387[7.42], R310[5.40] and W306[5.36]. However, with the exception of R310[5.40], where it forms a more persistent hydrogen bond, these interactions are again very transient and there are no interactions with TM3. Beyond these residues, there are very few interactions formed with the remainder of the N-terminal 9 residues of this peptide (Supplementary Table 3). Therefore, overall exendin-P5 forms relatively stable interactions with the ECD, the upper portion of TM1, TM2 and TM7, and with ECL1 and ECL2, however interactions with key residues deeper in TM2, TM3, TM5, TM6, TM7, as well as with ECL3, are transient, the majority of them present for <20% of the frames measured within the MD simulation (Supplementary Fig. 11, Supplementary Table 3). These data are consistent with alanine substitution of TMD residues lining the binding cavity generally having limited impact on exendin-P5 affinity, suggesting that its affinity is largely driven by interactions with the ECD (Figs. 4 and 5, Supplementary Fig. 9). In contrast, transient interactions between the exendin-P5 N-terminus and TMD are clearly important for agonism, with the

majority of residues within this cavity being required for eliciting cAMP signalling (Figs. 4 and 5, Supplementary Fig. 10).

**Dynamics of peptide–TMD interactions are correlated with the allosteric effect of G proteins on agonist affinity and G protein conformation.** The stark contrast in the requirement for stable TMD interactions for exendin-P5 affinity relative to GLP-1, oxyntomodulin and exendin-4, raises important questions regarding molecular mechanisms for peptide binding and receptor activation. To interrogate this, the influence of the bound G protein on the affinity of each agonist was assessed using wildtype and CRISPR-engineered HEK293 cells, where all Gα subtypes are depleted (Δall Gα HEK293)[34]. A NanoBRET membrane competition binding assay was employed to assess the ability of each peptide to inhibit binding of the fluorescent probe ROX-Ex4 to the GLP-1R N-terminally tagged with nanoluciferase (Nluc). In the wildtype cell line, GLP-1, oxyntomodulin and exendin-4 competition curves were clearly biphasic, with potencies for the high-affinity site correlating with those reported from whole cell-binding assays in the wildtype GLP-1R expressing ChoFlpIn cell line used in the mutagenesis study (Fig. 7a, Supplementary Figs. 5 and 6). In contrast, exendin-P5 exhibited monophasic binding curves with a lower pIC$_{50}$ than the other peptides that were consistent with the pIC$_{50}$ achieved in the ChoFlpIn whole-cell assay. While there was a small reduction in the pIC$_{50}$ for exendin-P5 in the Δall Gα cell line, this effect was relatively minor. In contrast, in the absence of Gα proteins, the high-affinity binding site for GLP-1, exendin-4 and oxyntomodulin were not observed; all displayed a single site for inhibition of Rox-Ex4 binding and with IC$_{50}$ values consistent with the lower affinity site in GLP-1R expressing membranes where Gα proteins were present (Fig. 7a). This loss of high-affinity binding could be reversed by the over-expression of G$_s$ in the Δall Gα cell line, consistent with G$_s$ allosterically influencing the affinity of GLP-1, exendin-4 and oxyntomodulin, with a much more limited effect on exendin-P5 (Fig. 7a). Interestingly, when G$_s$ was overexpressed, there was a larger influence on GLP-1 that exhibited the highest stability of interactions within the TMD binding cavity in the MD studies, compared to oxyntomodulin and exendin-4, which were more dynamic (Fig. 7a, Supplementary Movie 1).

A recent study revealed that while class A and class B1 GPCRs have similar kinetics for G protein recruitment, class B1 GPCRs exhibited slower G$_s$ turnover associated with slower nucleotide exchange and slower GTP hydrolysis[21]. Collectively this manifested as slower ligand-induced dissociation of Gα and Gβγ when assessed in whole cells using a G$_s$ Nanobit complementation assay. To assess if there is any potential for different GLP-1R agonists to display differences in G$_s$ turnover, we employed this same Nanobit G$_s$ complementation assay, to determine ligand-induced G protein dissociation in whole cells (Fig. 7b). In addition, we measured the kinetics of an earlier step in the G protein activation cycle using an assay to measure the G$_s$ conformational change upon coupling to the ligand-activated receptor (Fig. 7c). This assay is sensitive to the positioning of the Gαs α-helical domain (AHD), relative to the Ras homology domain (RHD) (performed in cell membranes in the absence of GTP), where separation of these domains is required for GDP release from the G protein[27,35,36]. Consistent with previous observations[13], we demonstrated that exendin-P5 exhibited faster G$_s$ conformational transitions, relative to GLP-1 and exendin-4, and this was coupled with faster dissociation of the G protein heterotrimer in the Nanobit assay (Fig. 7b, c). Oxyntomodulin also displayed significantly faster kinetics relative to GLP-1 in both assays, whereas exendin-4 was more similar to GLP-1. In the G$_s$ conformational assay, oxyntomodulin and exendin-P5 also exhibited a slightly lower maximal change in BRET than GLP-1 and exendin-4

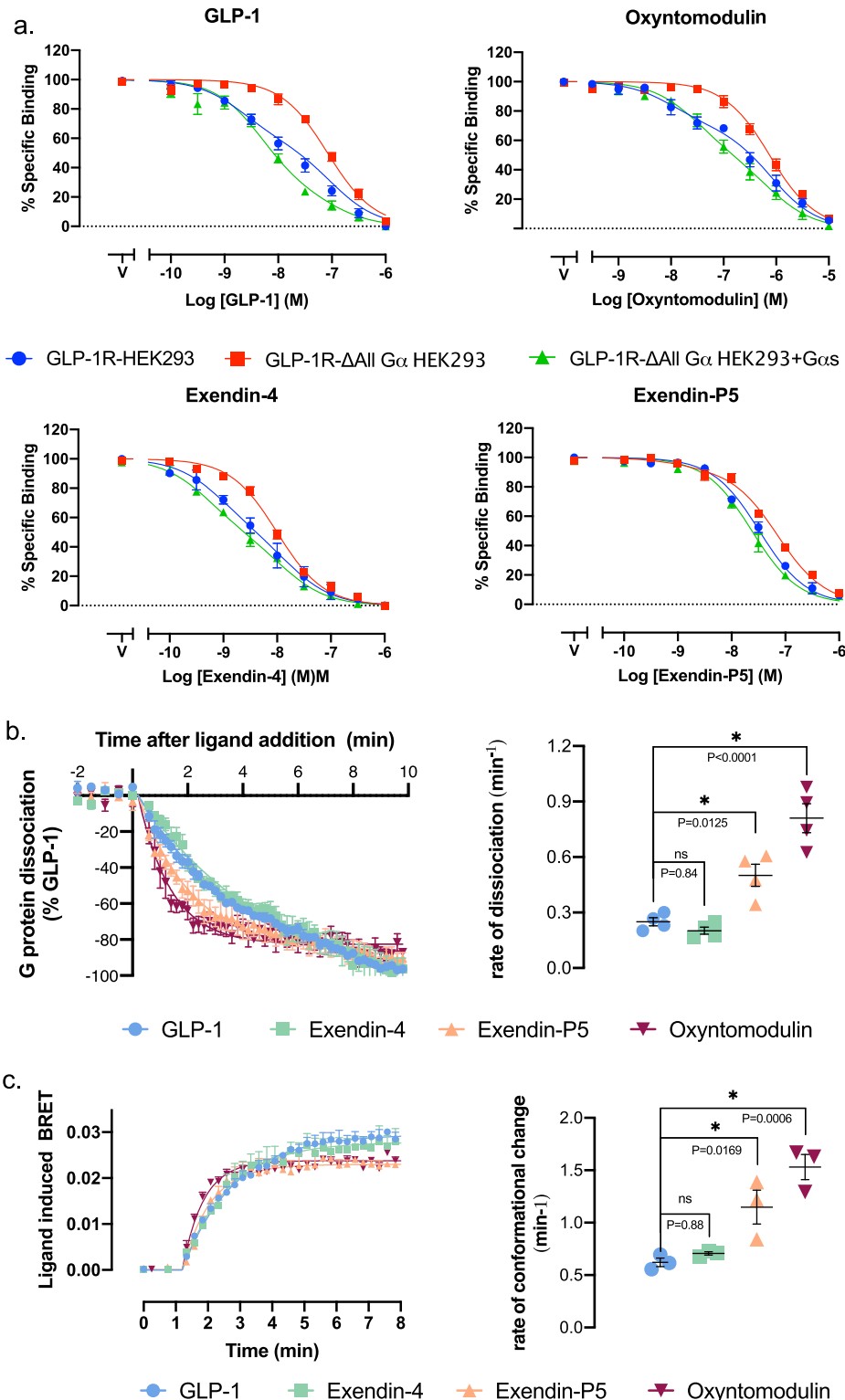

(Fig. 7c), suggesting a different ensemble of conformations of the AHD relative to the RHD, when bound by the different agonists.

**GLP-1R interactions with $G_s$ are conserved, however, MD simulations reveal ligand-specific effects on interaction dynamics.** Overlay of the four consensus cryo-EM static structures revealed very similar backbone conformations of the intracellular face of the receptor, with the greatest divergence for the ICLs (where modelled), and similar engagement with $G_s$ in all four peptides bound structures (Fig. 8). The C-terminal region of the α5 helix of $Gα_s$ was equivalently positioned for all structures, however, there was divergence at the N-terminal region of the α5, and this was translated across the remainder of the G protein

**Fig. 7 Allosteric effect of the G protein on peptide affinity and the peptide on $G_s$ activation. a** Equilibrium competition binding assays assessing the ability of GLP-1, oxyntomodulin, exendin-4 and exendin-P5 to compete for the probe Rox-Ex4, in HEK293 cells overexpressing the GLP-1R in the presence of endogenous Gα proteins (blue), the absence of Gα proteins (red) and when Gαs is overexpressed (no endogenous Gα proteins) (green). Data are presented as % specific binding with 100% binding defined as total probe binding in the absence of competing ligand and non-specific (0%) binding determined as probe binding in the presence of 1 μM exendin-4. Data are means ± s.e.m. of 7 independent experiments performed in duplicate. **b** HEK293A cells are transiently transfected with the GLP-1R and the NanoBit constructs for Gα$_s$ (Gα-Lgbit, Gγ$_2$-Smbit). Left; Luminescence signal was assessed over time (0–20 min) in the presence of saturating concentrations of GLP-1 (1 μM), exendin-4 (1 μM), oxyntomodulin (10 μM) and exendin-P5 (10 μM) and responses were normalised to the max loss of luminescence observed with GLP-1. Data shown are mean ± s.e.m. of four independent experiments performed in triplicate. Right; Quantification of the rate of G protein dissociation (luminescence change) for each agonist was calculated by applying a one-phase decay curve to the kinetic data with values from each individual experiment shown in circles with the mean ± s.e.m. of the four individual experiments. **c** Agonist-induced changes in trimeric $G_s$ protein conformation. Left; Ligand-induced changes in BRET were measured in plasma membrane preparations performed in a kinetic mode in the presence of saturating concentrations of GLP-1 (1 μM), exendin-4 (1 μM), oxyntomodulin (10 μM) and exendin-P5 (10 μM). Data shown are mean ± s.e.m. of three independent experiments performed in triplicate. Right; Quantification of the rate of ligand-induced conformational change for each agonist was calculated by applying a one-phase association curve to the kinetic data with values from each individual experiment shown in circles with the mean ± s.e.m. of the four individual experiments. * Represents statistically different to GLP-1 ($P < 0.05$) when assessed using a one-way ANOVA of variance with a Dunnett's post hoc test. Exact $P$ values are shown on the relevant figures. Source data are provided in the Source Data file.

(including the Gβγ subunits). The MD simulations described above included the $G_s$ heterotrimer and revealed that the receptor–$G_s$ interactions were very similar regardless of the bound agonist, albeit the majority of interactions were transient (Fig. 8b, Supplementary Fig. 13, Supplementary Tables 4 and 5). Nonetheless, when exendin-4, oxyntomodulin and exendin-P5 were bound, the receptor exhibited less persistent hydrogen bonding with the G protein when compared to GLP-1. In addition, each complex displayed transient van der Waals interactions between the Gα H4 and S6 domains and ICL3/TM6 of the receptor, that were, for the most part, not observed in the GLP-1 bound complex, albeit that specific interactions also differed between the individual peptide complexes (Fig. 8b, Supplementary Fig. 13, Supplementary Table 4). Moreover, while all the complexes displayed common and extensive interactions between the GLP-1R and the α5 helix of Gα$_s$, the persistence of interactions with individual residues differed in the exendin-P5 and oxyntomodulin bound complex (and to a lesser extent exendin-4), relative to GLP-1 over the time-course of the simulation (Fig. 8b, Supplementary Table 4). Both the exendin-P5 and oxyntomodulin bound complexes also exhibited less persistent interactions between ICL2 and the αN and hns1/S1 region of Gα$_s$ and between the N-terminal portion of H8 with G$_β$, whereas the remainder of interactions were largely consistent across the different complexes (Supplementary Fig. 13, Supplementary Tables 4, 5).

**Peptide agonists differentially engage receptor networks resulting in differences in the efficiency of communication.** To analyse the allosteric transmission of a signal from the peptide to G protein binding site, the MD simulations for each complex were analysed using Network and Community Analysis[37]. Correlation analysis revealed similar patterns in the presence of all peptides with highly correlated motions between residues within the ECD and within the TM domain, however, anti-correlated motions were evident between these two domains (Supplementary Fig. 14). In addition, there were correlated motions between the intracellular regions of the TM domain and the C-terminus of the G protein for all complexes, consistent with communication between the receptor and the G protein α5 helix. Cluster analysis comparing intramolecular interactions between side-chain residues in the GLP-1R TM bundle that was present for >75% of the MD simulation in at least one system revealed conserved interactions that were used by all peptides, consistent with a conserved mechanism of signal transmission from the peptide binding site to the G protein binding site (Supplementary Fig. 15). However,

the persistence of interactions differed suggesting subtle distinctions in how the different agonists engage these networks.

Correlation/cluster analysis alone is not sufficient to assess critical communication pathway(s) through the receptor, however, it forms the basis of the Network and Community Analysis, where a network is formed by an ensemble of nodes (residues) that are interconnected by edges, with an edge being a pair of critical nodes. In this analysis, edges connect non-consecutive nodes if the corresponding residues are within 4.5 Å for at least 75% of the MD frames. Variability in the connectivity of the networks enables the network to be subdivided into local communities according to the Girvan–Newman algorithm[38]. These communities contain groups of nodes that are densely interconnected and communicate to the rest of the network through a few (largely conserved) edges. Accordingly, nodes within the same community communicate with each other easily through multiple routes, whereas communication between critical nodes that cross the edges form bottlenecks for information transfer within the network. Application of the Girvan–Newman algorithm to the GLP-1R data splits the network into 14–18 communities for each complex, depending on the bound peptide (Supplementary Table 6), which includes 4–5 communities within the TMD. The analysis revealed differences in the number and the location of critical nodes required to communicate signals through the receptor by different peptides (Fig. 9, Supplementary Fig. 16). However, while these critical nodes for each peptide differ, all four peptides engage highly conserved class B1 GPCR residues, consistent with conserved activation mechanisms for this subclass of GPCRs.

Overall, more critical nodes are utilised by GLP-1 suggesting more effective communication between the G protein and the peptide-binding sites, relative to the other peptides, which is consistent with a greater allosteric influence of the G protein on GLP-1 affinity. In comparison, fewer conserved nodes were identified for communication between communities for the other peptides, particularly for oxyntomodulin (Supplementary Fig. 16). Interestingly, while this peptide engages fewer conserved nodes, overall the correlated motions within the receptor interactions were stronger than GLP-1 (Supplementary Fig. 14). Fewer critical nodes, particularly below the peptide binding site may suggest that oxyntomodulin has a shorter, therefore quicker, path of communication between the peptide and G protein binding site, relative to GLP-1. This is consistent with the faster G protein conformational change and activation exhibited by oxyntomodulin. Coupled with data from the interaction cluster analysis, which suggests similar patterns of interactions (with different

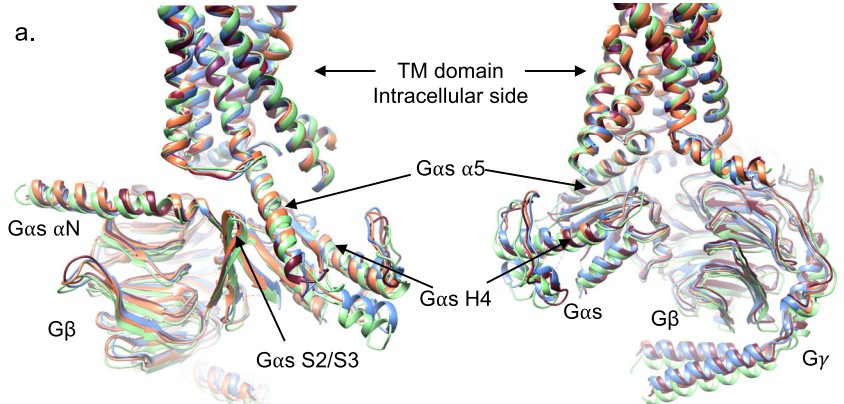

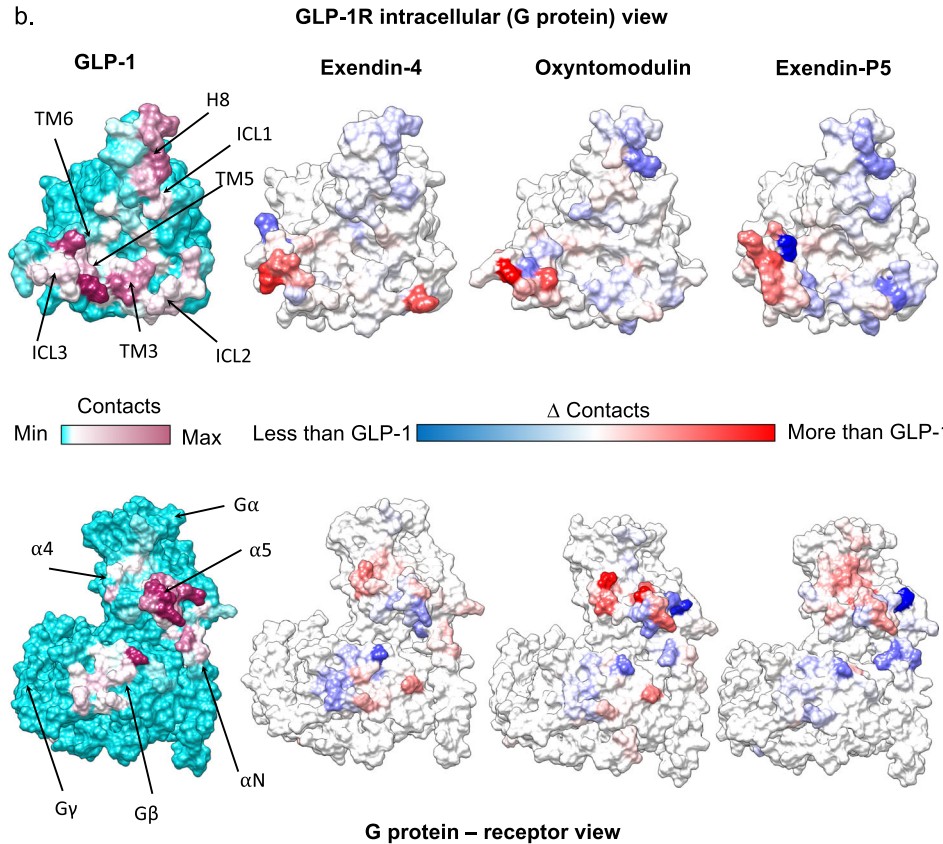

**Fig. 8 Interaction of G$_s$ with the GLP-1R in the presence of different peptide agonists. a** Superimposition of GLP-1R:G$_s$ structures bound with GLP-1 (PDB – 6X18, blue), exendin-4 (pale green), oxyntomodulin (dark pink) and exendin-P5 (pale orange) (PDB - 6B3J), viewing the GLP-1R interface. **b** Contact differences between the four complexes plotted on the receptor (top) and G$_s$ (bottom) surface determined from MD simulations. The first column shows the contacts between GLP-1R (top) and G$_s$ (bottom) during the MD simulations in the presence of GLP-1, with no contacts in cyan and increasing contacts heat mapped from white to dark pink. The other three columns report the contact differences (relative to GLP-1) for each residue of the GLP-1R and G$_s$ during the MD performed in the presence of the other agonists with blue indicating fewer contacts, white similar contacts, and red enhanced contacts.

occupancies) within the TMs, oxyntomodulin, may also potentially use multiple paths for communication of signal, rather than one, which could also explain the presence of fewer critical nodes for signal transmission (as the edge analysis considers nodes in proximity >75% of the time).

In contrast to the other peptides, exendin-P5 exhibits less correlated motions within the TM bundle (Supplementary Fig. 14) and requires fewer critical conserved nodes in the top half of the TM bundle (Supplementary Fig. 16), consistent with its very transient interactions, which likely allows it to communicate to

the TM bundle through multiple paths. However, while still requiring fewer conserved nodes in the base of the bundle than GLP-1, relative to oxyntomodulin more conserved nodes are used by exendin-P5 below the peptide-binding pocket to enable transmission of signalling to the G protein-binding site. This is again consistent with exendin-P5 exhibiting slower G protein activation kinetics than oxyntomodulin, yet faster than GLP-1. This analysis highlights the complexity of transmission of information within the receptor, with differences in the persistence of peptide–receptor interactions and receptor

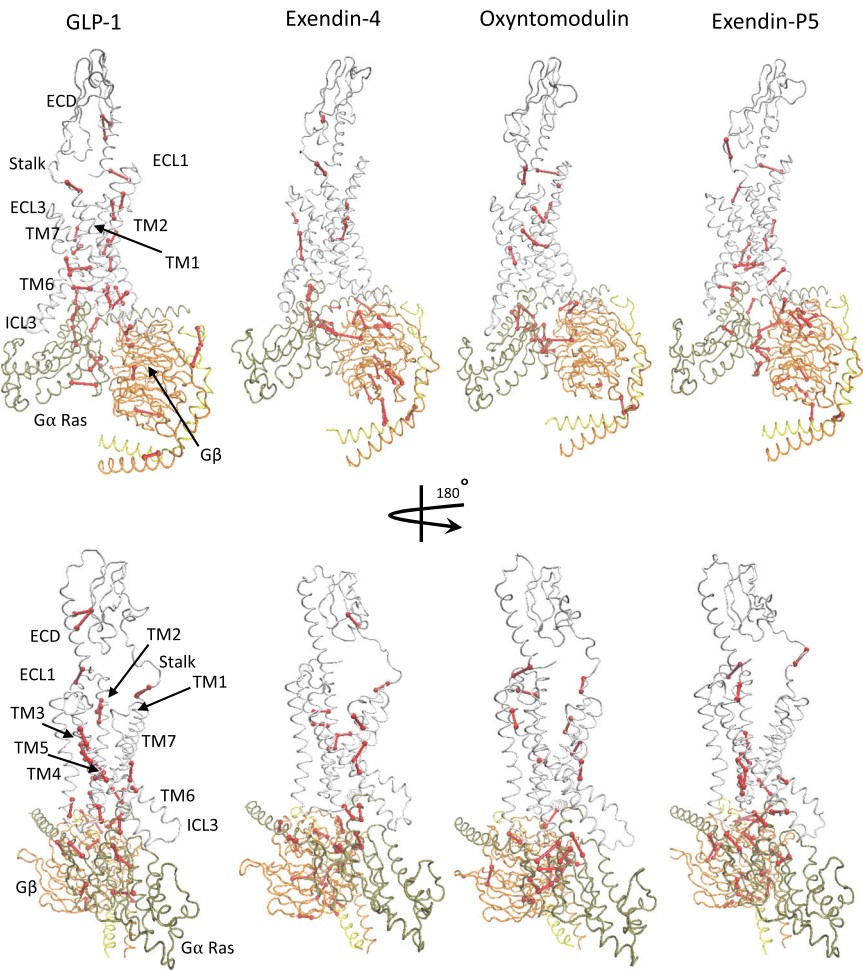

**Fig. 9 Position of critical nodes within GLP-1R, in complex with GLP-1, exendin-4, oxyntomodulin, or exendin-P5.** GLP-1R is in white ribbon, Gα in tan ribbon, Gβ in orange ribbon, and Gγ in yellow ribbon. Nodes and edges are in red. Peptides are not displayed as these were not considered during network analysis.

interactions linking the peptide and G protein-binding sites resulting in different efficacy (and bias) of different peptides agonists.

## Discussion

Combining experimentally determined GLP-1R structures with structure–function studies and simulations of receptor dynamics provides unique insights into how distinct agonists engage and activate the receptor. While GLP-1R peptide agonists bind both the ECD and the TMD, the role of the interactions with each domain differs among peptides. GLP-1, oxyntomodulin, and the clinically used mimetic exendin-4 are among the most extensively studied GLP-1R peptide agonists in functional and structure–function studies. Here, we reveal that while the peptides form similar interactions in the fully active, $G_s$-coupled state of the receptor, exendin-4 and oxyntomodulin-occupied receptors are more dynamic, which may, in part, be linked to the more transient nature of their interactions with the TMD observed in MD simulations. Oxyntomodulin is a biased agonist relative to GLP-1[4], and forms distinct and more dynamic interactions with the GLP-1R, particularly with residues at the base of the peptide binding cavity that is located above the conserved central polar network that is important for receptor activation. While the profile of exendin-4-mediated signalling is more similar to that of GLP-1, there are differences in receptor engagement by the two peptides that can be rationalised by

the structural and dynamic data. Like oxyntomodulin, exendin-4 exhibits more transient interactions than GLP-1 with residues at the base of the peptide binding cavity, albeit it that the pattern of interactions with the polar core are relatively conserved.

The peptide and G protein binding sites within GPCRs are allosterically linked to enable the transmission of information from peptide binding to G protein coupling. Network and Community analysis revealed the involvement of highly conserved class B1 GPCR residues for transmission of information through the GLP-1R TM bundle. While there was also conservation in the TM bundle interactions when engaged by the four peptides, differences were identified in how each peptide uses different networks to facilitate G protein coupling. This transmission of information across the TM bundle also enables G proteins to allosterically influence ligand affinity for GPCRs[39]. Consistent with this, we show the G protein can allosterically influence GLP-1R agonist affinity, but this occurs in a peptide-dependent manner. Enhanced affinity in the presence of $G_s$ is correlated with the degree of closure of the extracellular side of the TMD cavity around the peptide N-terminus and also the dynamics of interactions with residues in this domain. GLP-1, oxyntomodulin and exendin-4, whose affinities are influenced by the presence of $G_s$, promote a more closed bundle relative to exendin-P5, whose affinity is less sensitive to the presence of $G_s$. N-terminally truncated GLP-1 and exendin-4 peptides lacking the first 8 amino acids (GLP-1(15–36) and exendin(9–39)), exhibit

similar affinities in published studies[31], to their corresponding full-length peptides in the absence of G protein, providing further support that the influence of $G_s$ on peptide affinity is predominantly related to interactions of the N-terminus with the TMD. While it is likely that $G_s$ has the potential to influence the affinity of all peptides that bind in the TMD cavity, the specific amino acid sequence of individual peptides impacts complementarity with receptor residues, influencing the stability of interactions and contributing to the degree of allosterically-facilitated TMD closure around the peptide. With the exception of interactions of Glu1 and Asp3 with polar residues deep in the bundle, the first 9 residues of exendin-P5 do not form stable interactions with the TMD. Consequently, the TMD is more open and interactions with the peptide-N-terminus play only a minor role in its overall affinity. Nonetheless, these transient interactions are clearly essential for receptor activation, with mutation of the majority of residues within the TMD cavity decreasing exendin-P5 efficacy, but not affinity.

Previously we revealed that the exendin-P5-bound GLP-1R induces faster $G_s$ conformational transitions (that are linked to nucleotide exchange) and induces faster cAMP production when compared to the GLP-1-bound receptor[13]. In this study we demonstrate that both oxyntomodulin and exendin-P5 induce faster $G_s$ conformational transitions and $G_s$ heterotrimer dissociation, suggesting that these agonists may induce faster turnover of $G_s$ than GLP-1 and exendin-4. Interestingly, this profile appears to be correlated to the strength and nature of interactions of these peptides with key polar residues at the base of the GLP-1R binding cavity. While GLP-1, and to a slightly lesser degree exendin-4, form very stable interactions with R190[2.60], Y152[1.47], Y241[3.41] and E364[6.53], exendin-P5, and in particular, oxyntomodulin, form much more transient interactions with these key residues. This is predominantly due to the side chain chemistry of the residue at the position equivalent to Glu9 of GLP-1. While glutamic acid is conserved in exendin-4, this is replaced by aspartic acid and glutamine in exendin-P5 and oxyntomodulin, respectively. As polar residues at these receptor locations are conserved across class B1 GPCRs and play a role in receptor activation for all receptors where studied[40–45], stable vs transient interactions formed by peptide residues may also be associated with peptide efficacy at other class B1 receptors.

Given cryo-EM structures of the GLP-1R in complex with $G_s$ are stabilised by trapping the nucleotide-free G protein on the ligand-activated receptor, it is not surprising that the $G_s$ interactions are similar across the four structures, even when bound by different agonists, nonetheless, the MD simulations revealed potential differences in the dynamics of these interactions. Of particular note are the weaker interactions between ICL2 with the $\alpha$N and hns1/S1 region of $G\alpha_s$ and between H8 and $G_\beta$, in addition to differences in interactions between ICL3/TM6 with the $G\alpha_s$ $\alpha$5 helix within the exendin-P5 and oxyntomodulin-bound complexes, compared to GLP-1. Interactions of G proteins with these receptor domains contribute to the separation of the G protein RHD and AHD, disruption of the P loop and the nucleotide-binding pocket, all of which contribute to the release of GDP, one of the key rate-limiting steps in G protein activation[36]. However, how and if predicted differences in dynamics of receptor G protein interactions correlate to the differences in the effect of the allosteric coupling of the G protein on TMD binding sites, or the rates of G protein activation are unclear and additional studies will be required to address this.

Using the glucagon receptor as an exemplar, Hilger et al., identified that activated class B1 GPCRs exhibit a very persistent active state receptor conformation after G protein dissociation[21]. This implies that the activated receptor is primed to activate multiple rounds of G protein coupling following dissociation of

the initial interacting transducer protein, and this is proposed to contribute to the sustained cAMP signalling following activation of class B1 GPCRs. Consistent with this, exendin-P5, which induces faster kinetics in G protein activation and cAMP production, exhibits higher cAMP efficacy than GLP-1 and exendin-4, as indicated by a greater $pEC_{50}/pIC_{50}$ ratio when comparing cAMP and binding studies[13]. Our data suggest exendin-P5 may induce a "looser" coupling between the ligand TMD and $G_s$ binding sites, potentially linked to the more short-lived interactions with TMD residues, resulting in higher efficacy due to faster $G_s$ conformational transitions associated with nucleotide exchange and faster $G_s$ dissociation. Overall, given the long-lived active receptor conformation, this would promote greater turnover of $G_s$ and production of cAMP over time. A similar phenomenon has been observed at the calcitonin receptor, another class B1 GPCR, where human calcitonin, with a fast off-rate, turns over G protein faster than salmon calcitonin, which has a slow off-rate, and this was related to differences in the residency time of the G protein on the receptor and to the sensitivity of the G protein to GTP[35].

Oxyntomodulin exhibited even faster kinetics for $G_s$ conformational transitions and subunit dissociation than those induced by exendin-P5, and this was also correlated with more transient interactions at the base of the binding cavity and TM2. However, in contrast, this ligand does not exhibit higher efficacy relative to GLP-1, as determined by comparing transduction ratios from cAMP signalling and their determined affinity measures (Supplementary Table 2). This highlights the complexity of GPCR activation, where downstream signalling is influenced by the interplay of multiple transducers that can interact with activated receptors (pleiotropic coupling), and can also be influenced by different trafficking profiles that alter the location of the receptor in the cell. Oxyntomodulin is a biased agonist relative to GLP-1[1,3] (and exendin-P5[26]), with a bias towards arrestin recruitment over cAMP production, which would compete for G protein interactions and may contribute to the lack of correlation between the faster $G_s$ dissociation and enhanced efficacy when assessing cAMP accumulation in whole cells over an extended timeframe. While the structural basis of GLP-1R biased agonism between arrestin recruitment and G protein pathways is still not clear and will require additional studies to decipher, there is growing evidence that biased agonism is associated with the conformation and dynamics of the TM6/ECL3/TM7/TM1 domain[3,14,16,27,29], and this is consistent with the data presented herein. While in the consensus cryo-EM structures, GLP-1, exendin-4 and oxyntomodulin exhibit similar TMD conformations, MD simulations revealed that both exendin-4- and oxyntomodulin- bound GLP-1Rs, which exhibit biased agonism towards arrestin recruitment, are more dynamic in this region, than complexes with GLP-1 bound. In contrast, other agonists that exhibit stable open TM6/ECL3/TM7/TM1 conformations, including exendin-P5, are correlated with a bias towards G protein-mediated signalling.

In summary, combining structural data from cryo-EM, receptor mutagenesis, pharmacological assays and MD simulations advances our understanding of peptide agonist engagement with the GLP-1R. As class B1 peptide agonists engage their receptors via a two-domain interaction, their efficacy for G protein-mediated signalling is influenced by multiple factors, including the nature of interactions with the ECD and TMD, contributing to both peptide affinity and how ligand–receptor interactions influence G protein binding, nucleotide exchange and G protein dissociation from the receptor. This study provides insight into how differential dynamics of peptide–ligand engagement with the GLP-1R TMD can promote differences in G protein-mediated signalling, improving molecular understanding

of mechanisms that contribute to ligand-dependent differential efficacy at the GLP-1R.

## Methods

**Insect cell expression.** HA-signal peptide-FLAG-3C-GLP-1R-3C-8×HIS[13], human DNGα_s[46], His_6-tagged human Gβ_1and Gγ_2 were expressed in Tni insect cells (Expression systems) using baculovirus as previously described. Cell cultures were grown in ESF 921 serum-free media (Expression Systems) to a density of 4 million cells/ml and then infected with three separate baculoviruses at a ratio of 2:2:1 for GLP-1R, DNGα_s and Gβ_1γ_2. The culture was harvested by centrifugation 60 h post-infection and the cell pellet was stored at −80 °C.

**Complex purification.** Cell pellet was thawed in 20 mM HEPES pH 7.4, 50 mM NaCl, 5 mM CaCl_s, 2 mM MgCl_2 supplemented with cOmplete Protease Inhibitor Cocktail tablets (Roche) and benzonase (Merk Millipore). Complex formation was initiated by the addition of 10 μM exendin-4 or 50 μM oxyntomodulin (China Peptides), Nb35–His (10 μg/mL) and apyrase (25 mU/mL, NEB); the suspension was incubated for 1 h at room temperature. The complex was solubilized from the membrane by 0.5% (w/v) lauryl maltose neopentyl glycol (LMNG, Anatrace) supplemented with 0.03% (w/v) cholesteryl hemisuccinate (CHS, Anatrace) for 1 h at 4 °C. Insoluble material was removed by centrifugation at 30,000 × g for 30 min and the solubilised complex was immobilised by batch binding to M1 anti-FLAG affinity resin in the presence of 5 mM CaCl_2. The resin was packed into a glass column and washed with 20 column volumes of 20 mM HEPES pH 7.4, 100 mM NaCl, 2 mM MgCl_2, 5 mM CaCl_2, 1 μM exendin-4 or 10 μM oxyntomodulin, 0.01% (w/v) LMNG and 0.0006% (w/v) CHS before bound material was eluted in buffer containing 5 mM EGTA and 0.1 mg/mL FLAG peptide. The complex was concentrated using an Amicon Ultra Centrifugal Filter (MWCO 100 kDa) and subjected to size-exclusion chromatography on a Superdex 200 Increase 10/300 column (GE Healthcare) that was pre-equilibrated with 20 mM HEPES pH 7.4, 100 mM NaCl, 2 mM MgCl_2, 1 μM exendin-4 or 10 μM oxyntomodulin, 0.01% (w/v) LMNG and 0.0006% (w/v) CHS to separate complex from contaminants. Eluted fractions consisting of receptor and G-protein complex were pooled and concentrated to 3-5 mg/mL. The complex samples were flash-frozen in liquid nitrogen and stored at −80 °C.

**SDS–PAGE and Western blot analysis.** Sample collected from size-exclusion chromatography was analysed by SDS–PAGE and Western blot. For SDS–PAGE, precast gradient TGX gels (Bio-Rad) were used. Gels were either stained by Instant Blue (Expedeon) or immediately transferred to PVDF membrane (Bio-Rad) at 100 V for 1 h. The proteins on the PVDF membrane were probed with two primary antibodies, rabbit anti-Gα_s C-18 antibody (cat. no. sc-383, Santa Cruz) against Gα_s subunit and mouse penta-His antibody (cat. no. 34660, QIAGEN) against His tags. The membrane was washed and incubated with secondary antibodies, 680RD goat anti-mouse and 800CW goat anti-rabbit (LI-COR). Bands were imaged using an infra-red imaging system (LI-COR Odyssey Imaging System).

**Preparation of vitrified specimen.** EM grids (Quantifoil, Großlöbichau, Germany, 200 mesh copper R1.2/1.3) were glow discharged for 30 s in high-pressure air using Harrick plasma cleaner (Harrick, Ithaca, NY). The sample was applied on the grid in the Vitrobot chamber (FEI Vitrobot Mark IV). The chamber of Vitrobot was set to 100% humidity at 4 °C. The sample was blotted for 5 s with a blot force of 20 and then plunged into propane–ethane mixture (37% ethane and 63% propane).

**Data acquisition.** Exendin-4: Data for the GLP1R:DNGs:exendin- complex was collected on a Titan Krios microscope operated at 300 kV (ThermoFisher Scientific equipped with a Gatan Quantum energy filter operating in Zero Loss mode with an energy slit with of 20 eV and a Gatan K2 Summit direct electron detector (Gatan). Movies were taken in EFTEM nanoprobe mode, with 50 μm C2 aperture and no objective aperture, at a magnified pixel size of 0.87 Å. Each movie comprised 48 frames with a total dose of 48 e−/Å2, the exposure time was 8 s with a dose rate of 7 e−/pix/s on the detector. EPU (ThermoFisher Scientific) was used to automate data collection which involved implementing beam-tilt to collect a 3 × 3 grid of holes. As the data collection was split between two different days, the data was split into 18 optics groups.

Oxyntomodulin: Data for the GLP-1R:DNGs:oxyntomodulin complex was collected on a Titan Krios microscope operated at 300 kV (ThermoFisher Scientific equipped with a Gatan Quantum energy filter and a Gatan K2 Summit direct electron detector (Gatan) and a Volta Phase Plate (ThermoFisher Scientific). Movies were taken in EFTEM nanoprobe mode, with 50 μm C2 aperture, at a magnified pixel size of 1.06 Å. Each movie comprised 50 frames with a total dose of 50 e−/Å2, the exposure time was 8 s with a dose rate of 7 e−/pix/s on the detector. Data acquisition was done using SerialEM software at −500 nm defocus[44].

**Data processing.** Exendin-4: 8816 movies were collected and subjected to motion correction using motioncor2[47]. CTF estimation was done using Gctf software[48] on the non-dose-weighted micrographs. The particles were picked from dose-weighted

and low-pass filtered micrographs using crYOLO automated picking routine[49]. The particles were extracted in RELION 3.0[50] using a box size of 256 pixels. 1.52M picked particles were subjected to rounds of 2D and 3D classification in order to obtain a homogenous set of projections. This led to 422k particles which were polished and had their CTF parameters re-refined in RELION. Further rounds of 2D and 3D classification yielded a final particle stack of 275.7k particles for final 3D refinement and further rounds of masked refinements to reveal details of more flexible regions of the protein. Final consensus refinement produced a structure resolved to 3.59 Å (FSC = 0.143, gold standard). The cryo-EM data collection, refinement and validation statistics are reported in Supplementary Data Table 6.

Oxyntomodulin: Data processing of oxyntomodulin: 2364 movies were collected and subjected to motion correction using motioncor2[47]. Contrast transfer function (CTF) estimation was done using Gctf software on the non-dose-weighted micrographs[48]. The particles were picked using gautomatch (developed by K. Zhang, MRC Laboratory of Molecular Biology, Cambridge, UK; http://www.mrc-lmb.cam.ac.uk/kzhang/Gautomatch/). An initial model was made using EMAN2[51] based on a few automatically picked micrographs and using the common-line approach. The particles were extracted in RELION 2.03[52] using a box size of 180 pixels. Picked particles (1,070,980) were subjected to two rounds of 3D classification with three classes. Particles (209,000) from the best looking class were subjected to 3D auto-refinement in RELION 2.03. The refined revealed the final structure at 3.3 Å resolution. The cryo-EM data collection, refinement and validation statistics are reported in Supplementary Data Table 6.

**Modelling.** The sequence corrected model of exendinP5-GLP-1R-Gs (PDB: 6B3J)[13] was used as the initial template and fit in the cryo-EM density maps in UCSF Chimera (v1.14) for both the exendin-4 and oxyntomodulin bound structures, followed by molecular dynamics flexible fitting (MDFF) simulation with nanoscale molecular dynamics (NAMD)[53]. The fitted models were further refined by rounds of manual model building in COOT[54] and real-space refinement, as implemented in the PHENIX software package[55]. The ECD and ECLs were modelled manually without ambiguity based on the ECD-focused map. The density of the ECD linker (E127ECD-P137 ECD), ECL1 (T207ECL1-Q213ECL1/L218ECL1) and ICL3 (N338ICL3-D344ICL3) regions of both exendin 4 and oxyntomodulin complexes were discontinuous and these sequences were omitted from the final models. The ECL3 was less resolved in the exendin-bound complex and residues from E373ECL3 to R376ECL3 were omitted.

**ChoFlpIn stable cell lines generation.** The wild-type (WT) and mutant cMyc-GLP-1R[4] constructs containing designed signal alanine mutation were integrated into CHOFlpIn cells using the FlpIn Gateway technology system (Invitrogen). Stable CHOFlpIn expression cell lines were selected using 600 μg/ml hygromyocin B, and maintained in DMEM supplemented with 5% (V/V) FBS (Invitrogen) at 37 °C in 5% CO_2.

**Whole-cell radioligand-binding assay.** Cells were seeded at a density of 30,000 cells/well into 96-well culture plates and incubated overnight in DMEM containing 5% FBS at 37 °C in 5% CO_2. Growth media was replaced with binding buffer [DMEM containing 25 mM HEPES and 0.1% (w/v) BSA] containing 0.1 nM [125I]-exendin(9–39) and increasing concentrations of unlabelled peptide agonists. Cells were incubated overnight at 4 °C, followed by three washes in ice-cold 1× PBS to remove unbound radioligand[56]. Cells were then solubilised in 0.1 M NaOH, and radioactivity was determined by gamma counting. For all experiments, nonspecific binding was defined by 1 μM exendin(9–39).

**cAMP accumulation assay.** CHOFlpIn WT GLP-1R or CHOFlpIn mutant GLP-1R cells were seeded at a density of 30,000 cells per well into a 96-well plate and incubated overnight at 37 °C in 5% CO_2. cAMP detection was using a Lance cAMP kit (PerkinElmer Life and Analytical Sciences), performed as previously described[6]. Growth media was replaced with stimulation buffer [phenol-free DMEM containing 0.1% (w/v) bovine serum albumin (BSA) and 1 mM 3-isobutyl-1-methyl-xanthine] and incubated for 1 h at 37 °C in 5% CO_2. Cells were stimulated with increasing concentrations of ligand, 100 μM forskolin or vehicle, and incubated for 30 min at 37 °C in 5% CO_2. The reaction was terminated by rapid removal of the ligand-containing buffer and addition of 50 μL of ice-cold 100% ethanol. After ethanol evaporation, 75 μL of lysis buffer [0.1% (w/v) BSA, 0.3% (v/v) Tween 20, and 5 mM HEPES, pH 7.4] was added, and 10 μL of lysate was transferred to a 384-well OptiPlate (PerkinElmer Life and Analytical Sciences). 5 μL of 1/100 dilution of the Alexa Fluor® 647-anti cAMP antibody solution in the Detection Buffer (PerkinElmer Life and Analytical Sciences) and 10 μL of 1/5000 dilution of Eu-W8044-labelled streptavidin and Biotin-cAMP in the Detection Buffer (PerkinElmer Life and Analytical Sciences) were added in reduced lighting conditions. Plates were incubated at room temperature for 2 h before measurement of the fluorescence using an EnVision Multimode Plate (Reader PerkinElmer Life and Analytical Sciences). All values were converted to cAMP concentration using cAMP standard curve performed parallel and data were subsequently normalised to the response of 100 μM forskolin in each cell line, and then normalised to the WT for each agonist.

**Cell surface expression**. Cell surface expression was detected using a cell surface ELISA to detect a double c-Myc epitope label incorporated with the N-terminal region of the GLP-1R constructs. FlpInCHO wildtype and mutant human GLP-1R cells were seeded at a density of $25 \times 10^4$ cells/well into 24-well culture plates and incubated overnight at 37 °C in 5% $CO_2$, washed three times in 1× PBS and fixed with 3.7% paraformaldehyde (PFA) at 4 °C for 15 min. Cell surface receptor detection was then performed as previously described. Cell surface receptor detection was performed using a mouse monoclonal (9E10) primary antibody [1:2000] to detect the c-myc tag, and a mouse raised IgG Horse Radish Peroxidase-linked secondary antibody [1:2000], both diluted in blocking solution [1× PBS containing 2% (w/v) BSA and 0.05% (w/v) Tween-20]. Peroxidase activity was then measured using Sigma*Fast* OPD tablets (Sigma) according to the manufacturer's instructions, and fluorescence was detected at an emission wavelength of 492 nm. Data were normalised to the basal fluorescence detected in FlpInCHO parental cells.

**NanoBRET membrane-binding assays**. HEK293A WT and HEK293A ΔGα cells were transiently transfected with Nluc-hGLP-1R or Nluc-hGLP-1R + Gα$_s$ + Gβ1γ2. 48 h post-transfection, cells were pelleted and resuspended (~1.5 g packed volume) into 5 mL membrane preparation buffer followed by homogenisation with a polytron homogeniser at 4 °C. The homogenate was loaded onto a stepped sucrose gradient (40%/35%/22.5%/10%/homogenate) and centrifuged at 100,000×$g$ for 2 h 30 min at 4 °C. The 22.5%/10% interface (~1.5 mL) was collected and diluted to 17 mL with membrane preparation buffer followed by centrifugation at 100,000×$g$ for 30 min at 4 °C. The final pellet was resuspended in ~250 μL membrane preparation buffer and stored at −80 °C. On the day of assay 2 μg per well of the cell membrane was incubated with furimazine (1:1000 dilution from stock) in assay buffer (1× HBSS, 10 mM HEPES, 0.1% (w/v) BSA, 1× P8340 protease inhibitor cocktail, 1 mM DTT and 0.1 mM PMSF, pH 7.4). Rox-Ex4 was used as the fluorescent ligand in the NanoBRET binding assay. Membranes were increasing concentrations of peptides and RoxEx4 for 30 min prior to measurement of the BRET signal between Nluc-hGLP-1R and Rox-Ex4. This was assessed using a PHERAstar (BMG LabTech) at 10 s intervals (25 °C). A $K_d$ concentration, (3.16 nM for HEK293 WT cells, and 10 nM for HEK293 ΔGα cells) of Rox-Ex4 was used. Data were corrected for baseline and vehicle (probe only) responses.

**G$_s$ conformational change**. HEK293A cells stably expressing the GLP-1R (tested and confirmed to be free from mycoplasma) were transfected with a 1:1:1 ratio of Gγ2:venus–Gα$_s$:nanoluc–Gβ1 using a standard PEI protocol (6:1 ratio PEI:DNA). Cells were incubated overnight at 37 °C in 5% $CO_2$ and pelleted. Enriched plasma membranes were prepared as described for the NanoBRET membrane binding assays. 5 μg per well of cell membrane was incubated with furimazine (1:1,000 dilution from stock) in assay buffer (1× HBSS, 10 mM HEPES, 0.1% (w/v) BSA, 1× P8340 protease inhibitor cocktail, 1 mM DTT and 0.1 mM PMSF, pH 7.4). The GLP-1R-induced BRET signal between Gα$_s$ and Gγ was measured at 30 °C using a PHERAstar instrument (BMG LabTech). Baseline BRET measurements were taken for 2 min before the addition of the vehicle or ligand. BRET was measured at 15 s intervals for a further 10 min. All assays were performed in a final volume of 100 μl. Data were corrected for baseline and vehicle control. The concentration-response curves were then plotted using the total area under the curve during the time of measurement post ligand addition.

**G$_s$ Nanobit complementation assays**. HEK293AWT cells stably expressing the hGLP-1R were transiently transfected with Gα-LgBIT, Gβ1, Gγ2-SmBIT (1:5:5) 48 h before the assays using standard PEI transfection protocol. Cells were then incubated with coelenterazine H (5 μM) in assay buffer (1× HBSS, 10 mM HEPES, 0.1% (w/v) BSA) for 1 h at room temperature. Luminance signals were measured using a CLariostar (BMG LabTech) at every 30 s intervals before, and every 15 s intervals after ligand addition (25 °C). Data were corrected to baseline and vehicle-treated samples.

**Pharmacological data analysis**. Pharmacological data were analysed using Prism 8 (GraphPad). Concentration-response binding and signalling data were analysed using the one-site binding inhibition and the three-parameter logistic equations in Graphpad prism, respectively. This enables the determination of pIC50, pEC50 and Emax values. cAMP accumulation concentration-response curves were also analysed using an operational model of agonism modified to directly estimate the ratio of $\tau/K_A$ using Eq. (1)[57].

$$Y = \frac{E_m \left(\frac{\tau}{K_A}\right)^n [A]^n}{[A] \left(\frac{\tau}{K_A}\right)^n \left(\frac{1[A]}{K_A}\right)^n} \qquad (1)$$

where $E_m$ represents the maximal stimulation of the system, $K_A$ is the agonist–receptor dissociation constant, in molar concentration, [A] is the molar concentration of ligand and $\tau$ is the operational measure of efficacy in the system, which incorporates signalling efficacy and receptor density. Derived $\tau/K_A$ values were corrected to cell surface expression, measured by ELISA, and errors were propagated from both $\tau/K_A$ and cell surface expression.

For rate analysis of G protein BRET assays, data were fitted to a one-phase association curve in Graphpad Prism. Normalised AUC for the indicated ligand concentrations was plotted as a concentration-response curve and fitted with a three-parameter logistic curve.

Statistical analysis was performed using a one-way analysis of variance and a Dunnett's post-test, and significance was accepted at $P < 0.05$.

**MD methods**. The missing loops in the cryo-EM structures were reconstructed using modeller or by molecular superposition as described elsewhere[27]. The four GLP-1R complexes were prepared for MD simulations with the CHARMM36 force field[58], employing in-house python htmd[59] and TCL (Tool Command Language) scripts. Hydrogen atoms were first added at a simulated pH of 7.0 by means of the pdb2pqr[60] and propka[61] software, and the protonation of titratable side chains was checked by visual inspection. Each system was superimposed on the GLP-1R coordinates retrieved from the OPM database[62] in order to correctly orient the receptor before it was inserted[63] in a rectangular 125 Å × 116 Å 1-palmitoyl-2-oleyl-sn-glycerol-3-phosphocholine (POPC) bilayer (previously built by using the VMD Membrane Builder plugin 1.1, Membrane Plugin, Version 1.1. at http://www.ks.uiuc.edu/Research/vmd/plugins/membrane/), removing the lipid molecules overlapping the receptor TMs bundle. TIP3P water molecules[64] were added to the simulation box (125 Å × 116 Å × 195 Å) by means of the VMD Solvate plugin 1.5 (Solvate Plugin, Version 1.5. at <http://www.ks.uiuc.edu/Research/vmd/plugins/solvate/). Overall charge neutrality was finally reached by adding Na$^+$/Cl$^-$ counter ions (final ionic concentration of 0.150 M), using the VMD Autoionize plugin 1.3 (Autoionize Plugin, Version 1.3. at http://www.ks.uiuc.edu/Research/vmd/plugins/autoionize/).

**Systems equilibration and MD settings**. Equilibration and MD productive simulations were computed using ACEMD. Isothermal–isobaric conditions (Berendsen barostat[65] with a target pressure 1 atm; Langevin thermostat[66] with a target temperature 300 K and damping of 1 ps$^{-1}$) were employed to equilibrate the systems through a multi-stage procedure (integration time step of 2 fs). First, clashes between lipid atoms were reduced through 3000 conjugate-gradient minimisation steps, then a 2 ns long MD simulation was run with a positional constraint of 1 kcal mol$^{-1}$ Å$^{-2}$ on protein and lipid phosphorus atoms. Successively, 20 ns of MD were performed constraining only the protein atoms. In the last equilibration stage, positional constraints were applied only to the protein backbone alpha carbons, for a further 60 ns.

Four 500 ns-long replicas were simulated for each complex (2 μs of total MD time). Productive trajectories were computed with an integration time step of 4 fs in the canonical ensemble (NVT) at 300 K, using a thermostat damping of 0.1 ps$^{-1}$ and the M-SHAKE algorithm[67] to constrain the bond lengths involving hydrogen atoms. The cutoff distance for electrostatic interactions was set at 9 Å, with a switching function applied beyond 7.5 Å. Long-range Coulomb interactions were handled using the particle mesh Ewald summation method (PME)[68] by setting the mesh spacing to 1.0 Å.

**MD analysis**. Atomic contacts were computed using VMD[13]. A contact was considered productive if the distance between two atoms was lower than 3.5 Å. Hydrogen bonds were quantified using GetContacts analysis tool (at https://getcontacts.github.io/), a donor–acceptor distance of 3.3 Å and an angle value of 150° was set as geometrical cut-offs. Supplementary Movie 1 was produced employing VMD[69] and avconv (at https://libav.org/avconv.html). Cluster analysis of the contacts between GLP-1R side chains (Supplementary Fig. 15) was performed using the GetContacts analysis tool (at https://getcontacts.github.io/).

**Network and community analysis**. The four 500 ns-long replicas simulated for each complex were merged (2 μs for each system), and a stride of 1 ns was applied. Network and Community Analyses[37](code and protocol available at http://faculty.scs.illinois.edu/schulten/software/networkTools/index.html) were performed within the VMD[69] environment, considering the alpha carbon atoms of GLP-1R, Gα, Gβ, and Gγ subunits as nodes (the peptide agonist was not included). A network is formed by an ensemble of nodes interconnected by edges. Edges connect pairs of (non-consecutive) nodes if the corresponding residues are in contact (within 4.5 Å), for at least 75% of the frames. The resulting dynamical network was weighted by considering the probability $w_{ij}$ of information transfer across the edge connecting two nodes $i$ and $j$; calculated using Eq. (2).

$$W_{ij} = - \log(|C_{ij}|) \qquad (2)$$

where $|C_{ij}|$ measures the correlation values of motion between the two residues during the simulation (Supplementary Fig. 14).

The whole network was subdivided into local communities according to the Girvan–Newman algorithm[38]. The Girvan-Newman algorithm detects the communities within a network (e.g. nodes that are tightly joined together) by iteratively evaluating the number of shorter paths between nodes and removing the edges involved in the highest number of these. As the edges joining two communities likely take part in a high number of paths (edge betweenness centrality), by removing these edges it is possible to separate the network communities from one another.

**Amino acid conservation**. Amino acid conservation was determined using the ConSurf webserver[70], which estimated the rate of residue mutation over class B1 GPCR sequences.

**Graphics**. Molecular graphics images were produced using the UCSF Chimera (v1.14) and ChimeraX packages from the Computer Graphics Laboratory, University of California, San Francisco (supported by NIH P41 RR-01081and R01-GM129325)[71,72]. ShinyCircos[73] was used to generate flare plots depicting the GLP-1R TM bundle contacts.

**Reporting summary**. Further information on research design is available in the Nature Research Reporting Summary linked to this article.

## Data availability

All data generated in this study are included within the manuscript, Supplementary Information, Source Data file or has been deposited in available databases. The MD trajectories have been deposited on Zenodo (https://zenodo.org/record/5226209). Atomic coordinates and the cryo-EM density map have been deposited in the Protein Data Bank (PDB) and the electron microscopy data bank (EMDB), respectively, under PDB accession numbers 7LLY and 7LLL, and the EMDB entry IDs EMD-23436 and EMD-23425. Atomic coordinates for additional structures (GLP-1 and exendin-P5) that were not determined in this study, but used as the starting structure for the MD simulations are available from the PDB under accession codes 6B3J and 6X18. Source data are provided with this paper.

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

## Acknowledgements

This work was supported by the National Health and Medical Research Council of Australia (NHMRC) (project grant 1126857, ideas grant 1184726 (D.W.), and programme grant 1150083 (P.M.S.)). P.M.S. is a Senior Principal Research Fellow (ID: 1154434) and D.W. a Senior Research Fellow of the NHMRC (ID: 1155302). R.D. was supported by Takeda Science Foundation 2019 Medical Research Grant and Japan Science and Technology Agency PRESTO (18069571). K.J.G., K.L. and P.Z. are Future Fellows of the Australian Research Council (ARC; K.J.G—FT170100392, K.L—FT160100075, P.Z—FT200100218). M.J.B. is funded by a Fellowship from the ARC Industrial Transformation Training Centre for Cryo-electron Microscopy of Membrane Proteins (IC200100052). C.A.R. is a Royal Society Industry Fellow. This work was supported by the Monash University Ramaciotti Centre for cryo-electron microscopy and the Monash University MASSIVE high-performance computing facility.

## Author contributions

G.D. and C.A.R. designed, performed and analysed the MD simulations; Y.-L.L. and X.Z. expressed and purified the complexes; M.K. and R.D. vitrified the samples; M.K and H.V imaged the samples to acquire EM data; X.Z. and M.K. processed the EM data and performed EM map calculations; X.Z, M.J.B and A.K. built the models and performed refinements; L.C. and T.T.T. performed the mutagenesis studies; P.Z. and T.T.T. performed the G protein and binding studies; G.D., X.Z., M.J.B., C.A.R., P.Z., P.M.S. and D.W. performed data analysis and data interpretation; Y.-L.L., A.G., K.J.G., K.L., A.C., R.D. assisted with data interpretation, and supervision for the project; P.Z., C.A.R., P.M.S. and D.W. designed and supervised the project. P.M.S. and D.W. wrote the manuscript with assistance from P.Z., G.D., C.A.R., X.Z. and M.J.B. All authors reviewed and revised the manuscript.

## Competing interests

The authors declare no competing interests.
