## [Peer Review File · Nature Communications]

Dynamics of GLP-1R peptide agonist engagement are correlated with kinetics of G protein activationREVIEWER COMMENTS

Reviewer #1 (Remarks to the Author):

In this study, Deganutti et al. reported two cryo-EM structures of Gs-coupled GLP-1R bound to exendin-4 and oxyntomodulin at the resolution of 3.3 and 3.7 angstroms, respectively. By combining cryo-EM structures with molecular dynamics, mutagenesis, and pharmacological analyses, the authors discussed the distinctions in the static and dynamic interaction modes of exendin-4, oxyntomodulin, GLP-1, and exendin-P5 with GLP-1R. These findings also revealed that the distinctions in peptide N-terminus and dynamics with the receptor TMD are associated with differences in the allosteric coupling to G proteins, thus linking the dynamics of GLP-1R peptide agonist engagement to the kinetics of G protein activation.

Comments:

1. Line 340-344: "Interestingly, single alanine amino acid substitutions.....". It is confusing for the conclusion that "the non-optical interactions with the TMD likely promote faster peptide dissociation from the ECD." Please explain or rewrite this sentence.
2. Line 236: "...along the peptide exposed face of TM2, ECL1 and ECL2." doesn't make much sense. The interactions of GLP-1 and oxyntomodulin with receptor ECL1 are similar according to supplemental Figure 11.
3. The TMs and ECLs of the receptor should be labeled in Figure 6A for a clear presentation.
4. Did the authors add cholesterol in the system setup process? If not, please explain it since cholesterol could have an allosteric modulatory role for GLP-1R.
5. The reviewer noticed that the systems experienced several minimization rounds. However, is the temperature of the systems set to 300K from the start of the MD simulation? A sudden high temperature may cause irrational contacts and make the system collapse. The authors should prove the feasibility of their protocol.

Reviewer #2 (Remarks to the Author):

In the present study, Deganutti et al performed structural biology, molecular dynamics simulations, receptor mutagenesis, and pharmacological studies to explore the different activation mechanisms of

GLP-1R by four peptide agonists, including glucagon-like peptide-1, oxyntomodulin, exendin-4 and exendin-P5. Overall, the experiments are well designed and extensive. However, I have several concerns regarding MD simulations. These issues should be addressed to strengthen the current study.

1) Nowadays, 500 ns time scale of MD simulations should not to say long simulation. Since GPCR is a membrane protein which is embedded into the lipid bilayer along the simulations, it requires more relation time for GPCR to reach equilibration and to transmit the agonist binding to the G protein. Thus, I suggest the authors to extend the simulation to several microseconds to re-examine the detailed GPCR-agonist interactions.

2) The authors performed MD simulations in a single trajectory. This is biased because multiple replicas of MD simulations may generate different conformational dynamics of proteins. Thus, to reproduce the results, the authors should run the four systems at least in three independent replicas with random velocities.

3) The communication pathway triggered by the agonist binding to the G protein should be pinpointed using network analysis. Is this pathway conserved or not by the four different agonists?

Reviewer #3 (Remarks to the Author):

I applaud the authors in the attempt to quantify the effects of ligands responsible for biased signaling; the usual idea is that differential conformations of the receptor are stabilized by different agonists to do this but not much more beyond that. Having said that, I am wary of papers that only seek to 'increase our understanding. Not to take a Philistine view, it would be good if the authors would try to use what they've (we've) learned to help pharmacologists and medicinal chemists make better therapeutic agonists. The idea that kinetics are affected should not be surprising since allostereism is just that....a given PAM or NAM alters the rate of onset and or offset of the ligand it modulates... substitute agonist for PAM and G protein for the ligand and you have the same thing. I have no substantive criticism of this work as the experiments seem to have been carried out well and this group is established as a top group in the field. However, it is groups like this who should take the lead in going beyond just exploring what is happening and apply it to real therapeutic problems. However, having said that I know that understanding is the first step to doing this so I cannot fault this paper for being an excellent first step

Response to reviewers comments

Reviewer #1

In this study, Deganutti et al. reported two cryo-EM structures of Gs-coupled GLP-1R bound to exendin-4 and oxyntomodulin at the resolution of 3.3 and 3.7 angstroms, respectively. By combining cryo-EM structures with molecular dynamics, mutagenesis, and pharmacological analyses, the authors discussed the distinctions in the static and dynamic interaction modes of exendin-4, oxyntomodulin, GLP-1, and exendin-P5 with GLP-1R. These findings also revealed that the distinctions in peptide N-terminus and dynamics with the receptor TMD are associated with differences in the allosteric coupling to G proteins, thus linking the dynamics of GLP-1R peptide agonist engagement to the kinetics of G protein activation.

Comments:

1. Line 340-344: "Interestingly, single alanine amino acid substitutions.....". It is confusing for the conclusion that "the non-optical interactions with the TMD likely promote faster peptide dissociation from the ECD." Please explain or rewrite this sentence.

Exendin(9-39) is identical in sequence to exendin-4 in its interactions with the GLP-1R ECD, but lacks 8 residues that interact with the TMD. At many of the TMD mutant receptors, exendin-4 is still able to activate the receptor, but with lower potency, therefore the peptide must still engage with the TMD. Despite this, the affinity of exendin-4 at many of the TMD single amino acid substitutions is lower than exendin(9-39) at the wildtype receptor, even though this peptide makes only limited interactions with the TMD. This implies that by changing single residues within the TMD, interactions of the first 8 amino acids of exendin-4 are altered and in turn influence the affinity of C-terminus of the peptide that binds predominantly to the ECD (in a similar manner to exendin(9-39)). To clarify the meaning of the sentence, this has been modified to the following (with changes from the previous version in blue).

"Interestingly, single alanine amino acid substitutions of some interacting residues within the TMD had much larger effects on exendin-4 affinity than removal of the first 8 residues of the peptide (exendin (9-39)), suggesting that peptide ECD and TMD interactions are correlated; non-optimal interactions of exendin-4 with the TMD elicited by receptor alanine mutations, likely promotes faster peptide dissociation from the ECD, compared to when the TMD interacting residues are not present in the peptide".

2. Line 236: "...along the peptide exposed face of TM2, ECL1 and ECL2." doesn't make much sense. The interactions of GLP-1 and oxyntomodulin with receptor ECL1 are similar according to supplemental Figure 11. We apologise that this was not clear. We were referring to the side of the TM2 helix that is facing the peptide. We also thank the reviewer for pointing out the interactions within ECL1 are similar according to Supp Fig. 11, with only minor differences in the overall persistency of interactions, albeit the exact nature of their interactions do differ (as shown in Supplemental Table 3). To address the reviewers comment we have removed the peptide exposed face and ECL1 from this sentence – it now reads

"with more transient contacts deep within the peptide binding cavity and with residues located higher within TM2, and with ECL2".

3. The TMs and ECLs of the receptor should be labeled in Figure 6A for a clear presentation. We thank the reviewer for the suggestion. However, these figures already have a lot of labels. For the extracellular views, the ECLs and TMs are labelled in the reference panel (GLP-1), with the receptors bound by other peptides shown in exactly the same orientation, therefore these should not require labels to identify them. For the side views, the labelling of receptor residues includes the class B numbering (which identifies the relevant TM or ECL in subscript). We believe this should be sufficient to orient locations of the key residues within the receptor binding site. We have not added additional labels, as this overcrowds the figure, as it already contains a lot of labels.

4. Did the authors add cholesterol in the system setup process? If not, please explain it since cholesterol could have an allosteric modulatory role for GLP-1R.

Cholesterol was not added to the POPC membrane model. The presence of cholesterol would increase the variability of the systems and create a bias. Some replicas could be influenced by cholesterol more than others. Obtaining a convergence of receptor-cholesterol interactions in all the simulations would require up to tens of microseconds due to the low diffusibility in the lipid medium.

5. The reviewer noticed that the systems experienced several minimization rounds. However, is the temperature of the systems set to 300K from the start of the MD simulation? A sudden high temperature may cause irrational contacts and make the system collapse. The authors should prove the feasibility of their protocol.

As correctly noticed by the reviewer, the target temperature was set to 300K from the start of the equilibration. Just one initial minimization round was performed before three stages of equilibration, during which different positional restraints were applied. The restraints preserved the protein structure during the initial stages of equilibration. This equilibration protocol has been employed in numerous publications and previously assessed in terms of protein stability, area per lipid, and simulation box volume convergence, on several occasions (the most recent: Bolcato G. *et al.* doi.org/10.3390/biom10050732).

Reviewer #2.

In the present study, Deganutti et al performed structural biology, molecular dynamics simulations, receptor mutagenesis, and pharmacological studies to explore the different activation mechanisms of GLP-1R by four peptide agonists, including glucagon-like peptide-1, oxyntomodulin, exendin-4 and exendin-P5. Overall, the experiments are well designed and extensive. We thank the reviewer for the positive review of our work.

However, I have several concerns regarding MD simulations. These issues should be addressed to strengthen the current study. We have addressed the reviewer's comments below.

1) Nowadays, 500 ns time scale of MD simulations should not to say long simulation. Since GPCR is a membrane protein which is embedded into the lipid bilayer along the simulations, it requires more relation time for GPCR to reach equilibration and to transmit the agonist binding to the G protein. Thus, I suggest the authors to extend the simulation to several microseconds to re-examine the detailed GPCR-agonist interactions.

We agree with the reviewer that a single 500ns long simulations is not a very extensive MD sampling. Indeed, we increased the quality of the sampling running four independent parallel replicas for each complex. Collecting more parallel replicas is usually better than a single simulation run for the same amount of time (Knapp P. *et al* - doi.org/10.1021/acs.jctc.8b00391). Thus, we collected four independent replicas (2 μ s) for each system. It should also be noted that the starting point for these simulations was the cryo-EM structure for each complex which is already coupled to agonist and G protein, therefore the “transmission” of agonist to G protein binding is already established in the structure (these are not added during the simulation).

2) The authors performed MD simulations in a single trajectory. This is biased because multiple replicas of MD simulations may generate different conformational dynamics of proteins. Thus, to reproduce the results, the authors should run the four systems at least in three independent replicas with random velocities. As aforementioned, in line with the reviewer’s suggestion, we performed four independent replicas (2 μ s in total) for each system, reassigning the velocities according to the Boltzmann distribution at 300K.

3) The communication pathway triggered by the agonist binding to the G protein should be pinpointed using network analysis. Is this pathway conserved or not by the four different agonists?. As suggested by the reviewer, we have now applied network and community analysis using the method outlined in Sethi *et al* (PNAS 2009), and while this is informative, it further highlights the complexity of allosteric communication through GPCRs. We have mapped the edge residues/nodes that connect different communities in the physical networks for more than 75 % of the MD frames for each complex. These results demonstrate high peptide dependency with the four peptides using different nodes to differing degrees, however they all engage highly conserved residues within the class B1 GPCR subfamily for transmission of signalling. The analysis highlights that the allosteric links through the receptor between the peptide binding site and G protein binding site is most efficient for GLP-1 as there are more linked nodes (for greater than 75% of the simulation), which is consistent with the greater enhancement in GLP-1 affinity mediated by the G protein for this peptide. Oxyntomodulin uses fewer conserved nodes than the other peptides but, relative to GLP-1, the correlated motions as a whole are much stronger (and to a lesser extent exendin-4), whereas they are weaker for exendin-P5 that formed fewer interactions with the binding site. However, while fewer persistent interconnected edge interactions between communities were present (>75% of the time) for exendin-4, exendin-P5 and oxyntomodulin relative to GLP-1, cluster analysis of the contacts between GLP-1R TM side chains for all four complexes where an interaction was observed >75 % in at least one of the systems revealed much more subtle differences in the residues used by the different ligands. This showed that the majority of interactions are conserved and used

by all peptides, which is consistent with a conserved signal transmission mechanism from the peptide binding site to the G protein binding site at the intracellular face of the receptor. However, the persistence of interactions differs and is consistent with differences in the critical nodes required for communication through the TM bundle identified in the community analysis, and the varying degrees of influence of the G protein on the affinities of the different peptides. This analysis, coupled with the other data in the manuscript highlights the complexity of transmission of information within the receptor, with subtle differences in the persistence of peptide-receptor interactions and receptor interactions linking the peptide binding site to G protein binding site leading to differences in the efficacy of different agonists. These studies have been incorporated into the manuscript (Figure 9, Supplemental Figures 14-16, Supplemental Table 6, the last section of Results, plus a small paragraph in the discussion).

Reviewer #3.

I applaud the authors in the attempt to quantify the effects of ligands responsible for biased signaling; the usual idea is that differential conformations of the receptor are stabilized by different agonists to do this but not much more beyond that. Having said that, I am wary of papers that only seek to 'increase our understanding. Not to take a Philistine view, it would be good if the authors would try to use what they've (we've) learned to help pharmacologists and medicinal chemists make better therapeutic agonists. The idea that kinetics are affected should not be surprising since allosterism is just that....a given PAM or NAM alters the rate of onset and or offset of the ligand it modulates... substitute agonist for PAM and G protein for the ligand and you have the same thing. I have no substantive criticism of this work as the experiments seem to have been carried out well and this group is established as a top group in the field. However, it is groups like this who should take the lead in going beyond just exploring what is happening and apply it to real therapeutic problems. However, having said that I know that understanding is the first step to doing this so I cannot fault this paper for being an excellent first step.

We thank the reviewer for their positive comments and agree that further work from ourselves and others in the field is required to link structural and pharmacological data to therapeutic outcomes. However, as also noted by the reviewer, this work is the first step in understanding how biased agonism arises at the molecular level, and further studies are far beyond the scope of the current manuscript.

REVIEWERS' COMMENTS

Reviewer #1 (Remarks to the Author):

The authors have addressed all the comments of the reviewer. The manuscript has also been well improved.

Reviewer #2 (Remarks to the Author):

I highly appreciate the authors in considering my comments in their revised manuscript. According to this revision, the manuscript has been strengthened. Thus, I recommend to publish this paper in its current form.